# UniUGG: Unified 3D Understanding and Generation via Geometric-Semantic Encoding

**Yueming Xu**[1*]  **Jiahui Zhang**[1*]  **Ze Huang**[1*]  **Yurui Chen**[1]  **Yanpeng Zhou**[2]  **Zhenyu Chen**[2]
**Yu-Jie Yuan**[2]  **Pengxiang Xia**[2]  **Guowei Huang**[2]  **Xinyue Cai**[2]  **Zhongang Qi**[2]  **Xingyue Quan**[2]
**Jianye Hao**[2]  **Hang Xu**[2]  **Li Zhang**[1†]

[1] Fudan University    [2] Huawei Noah's Ark Lab

https://fudan-zvg.github.io/UniUGG

## Abstract

Despite the impressive progress on understanding and generating images shown by the recent unified architectures, the integration of 3D tasks remains challenging and largely unexplored. In this paper, we introduce *UniUGG*, the first unified understanding and generation framework for 3D modalities. Our unified framework employs an LLM to comprehend and decode sentences and 3D representations. At its core, we propose a spatial decoder leveraging a latent diffusion model to generate high-quality 3D representations. This allows for the generation and imagination of 3D scenes based on a reference image and an arbitrary view transformation, while remaining supports for spatial visual question answering (VQA) tasks. Additionally, we propose a geometric-semantic learning strategy to pretrain the vision encoder. This design jointly captures the input's semantic and geometric cues, enhancing both spatial understanding and generation. Extensive experimental results demonstrate the superiority of our method in visual representation, spatial understanding, and 3D generation.

## 1 Introduction

Recent work on unified 2D understanding and generation has made significant strides (Sun et al., 2023; 2024; Ye et al., 2024; Dong et al., 2024; Wu et al., 2024; Team, 2024; Wang et al., 2024a; Liu et al., 2024a; Wu et al., 2025; Chen et al., 2025; Huang et al., 2025). Early works (Sun et al., 2023; 2024; Ye et al., 2024; Dong et al., 2024) build unified frameworks that couple an autoregressive LLM with a diffusion image decoder. The LLM consumes text–vision inputs and produces a fixed set of learnable queries whose features are regressed into the diffusion latent space; the diffusion model then synthesizes the image conditioned on these latents. Subsequent works (Team, 2024; Liu et al., 2024a; Wu et al., 2024) employ VQ tokenizers for the unified generation of texts and images.

Although the aforementioned works have made significant progress in images, the challenge of achieving unified understanding and generation for 3D modalities remains largely unexplored. Several benchmarks (Zhang et al., 2025b; Fu et al., 2024b; Ma et al., 2024; Yang et al., 2025) integrate large-scale public datasets and design spatial VQA tasks to enhance the spatial reasoning capability of LLMs. However, these efforts focus solely on spatial understanding and take a rather brute-force approach—directly fine-tuning LLMs with large amounts of spatial data, which has shown limited effectiveness. Other works (Chen et al., 2024a; Cheng et al., 2024; Hong et al., 2023; Driess et al., 2023; Chen et al., 2024e; Zhu et al., 2024a) incorporate additional modalities, such as depth, point clouds, or scene graphs, to handle spatial understanding. Unfortunately, these methods introduce additional disadvantages, as they often require specialized data acquisition devices or necessitate explicit spatial modeling of the entire scene. These drawbacks hinder potential development for practical applications.

We identify two main challenges that bottleneck progress for the unified 3D frameworks. **The first issue is the limitation of visual representations.** Current LLMs typically rely on vision encoders

---

* Equally contributed    † Corresponding author (lizhangfd@fudan.edu.cn)

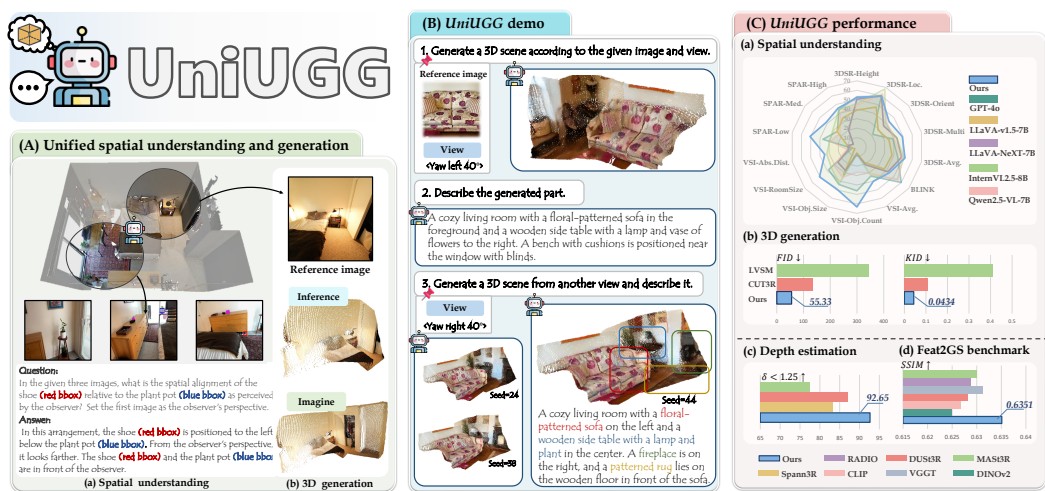

Figure 1: **We introduce *UniUGG*, the first unified framework for spatial understanding and generation.** (A) *UniUGG* supports spatial-level VQA and generates geometrically consistent 3D scenes. (B) Given a reference image, it can creatively generate 3D variations and describe them accurately. (C) *UniUGG* outperforms baselines in both spatial understanding and generation, with our specially tuned vision encoder excelling in downstream tasks.

pretrained on 2D image semantic tasks, which lack the capability to model 3D geometries. This limitation creates a performance bottleneck, particularly in spatial understanding tasks. **The second issue is the incompatibility between 3D generation and LLMs.** Here, LLMs are built on the basis of tokenization methods to autoregressively generate the next token. This scheme adapts well to image generation, as images are regular and can be tokenized into a fixed number of elements. However, such tokenization approaches are not easily applicable to 3D data such as point clouds due to the irregular nature of representations. This tokenization gap makes it more challenging for LLMs to handle autoregressive 3D generation tasks effectively.

Recently, a line of works (Wang & Agapito, 2024; Leroy et al., 2024; Zhang et al., 2024; Wang et al., 2025b;a) initiated by DUSt3R (Wang et al., 2024b) have introduced a new perspective on spatial representation by aligning pixels across multi-view into a unified global coordinate system. The multi-view geometry training paradigm enables models to reconstruct 3D scenes from visual inputs, along with predicting spatial relationships. Inspired by this, we introduce *UniUGG*, the first framework for unified spatial understanding and generation, marking a significant step by solving the aforementioned two issues.

For the first issue, we design a geometric-semantic learning strategy for vision encoder pretraining. This strategy incorporates semantic information from a teacher model while integrating the encoder with a spatial decoder for end-to-end multi-view geometric training, thereby enhancing its spatial modeling capabilities. The resulting ViT representations significantly improve both understanding and generation within the unified framework and yield better results in a variety of downstream tasks. The learning strategy not only provides the LLM with an enriched representation but also bridges the gap between vision and 3D using the spatial decoder. This decoder, a byproduct of the pretraining process, decodes visual representations into 3D scenes corresponding to the two inputs. Upon these, we solve the second issue by designing *UniUGG*. *UniUGG* takes a reference visual representation and an encoded target-view raymap as input, producing conditional features. These features are then used with a diffusion model to generate the visual representation of the target-view. To enhance this process, we design the Spatial-VAE, which effectively compresses geometric-semantic information, enabling more accurate and efficient representation generation. Additionally, it links the spatial decoder for end-to-end fine-tuning, enhancing information compression while mitigating the negative impact of discrepancies between the reconstructed and original representations on 3D scene decoding. Finally, both the original and generated visual representations are passed through the fine-tuned spatial decoder to decode the 3D scene. Thanks to the LLM-based architecture, *UniUGG* simultaneously learn understanding and generation tasks, enabling its 3D scene inference, while maintaining spatial VQA capabilities for both real and generated representations.

Our main **contributions** are summarized as follows: **(i)** We propose the first LLM-based unified generation and understanding framework for 3D scenes, *UniUGG*, which enables spatial-level VQA and generates geometrically consistent rich 3D environments. **(ii)** We introduce a novel geometric-semantic vision encoder pretraining strategy. Here, our ViT encodes geometric cues from input image pairs and preserves semantic features from 2D priors. **(iii)** We present a Spatial-VAE as the core of our 3D scene representation generation scheme. Our Spatial-VAE compresses the 3D geometric-semantic representations from input image pairs and helps producing sharper 3D point clouds as output. **(iv)** Our method achieves top performance on multiple spatial reasoning benchmarks, surpassing baselines on VSI-Bench by 17.9% in particular, and maintaining significant superiority in 3D generation.

## 2 RELATED WORKS

**Language models for spatial reasoning and generation**   There has been growing interest in applying large multimodal language models to spatial reasoning tasks. Recent models (Alayrac et al., 2022; Driess et al., 2023; Liu et al., 2023; Li et al., 2024a; Bai et al., 2023) have shown impressive results in language-guided visual understanding, but they often focus primarily on semantic alignment. As a result, they exhibit clear limitations when it comes to understanding spatial relations, viewpoint changes, and structural consistency. Several benchmarks (Zhang et al., 2025b; Fu et al., 2024b; Ma et al., 2024; Yang et al., 2025) have been proposed to evaluate spatial reasoning abilities, and existing methods (Chen et al., 2024a; Cheng et al., 2024; Hong et al., 2023; Driess et al., 2023; Chen et al., 2024e; Zhu et al., 2024a) typically enhance performance by increasing training data or incorporating additional structural inputs, such as depth, point clouds, or scene graphs. However, these structural inputs are often used without explicit modeling of spatial consistency, and structure-aware visual representations remain underexplored. In addition, while recent works have achieved unified 2D understanding and generation (Sun et al., 2023; 2024; Ye et al., 2024; Dong et al., 2024; Wu et al., 2024; Team, 2024; Wang et al., 2024a; Liu et al., 2024a; Wu et al., 2025; Chen et al., 2025; Huang et al., 2025), there is limited research on applying this concept to the spatial level. In contrast, we propose the first unified spatial framework, which not only handles spatial reasoning tasks but also generates 3D scenes based on a reference image and a specified view transformation.

**Semantic and geometric representation**   Vision encoders (Liang et al., 2024; Zhai et al., 2023; Cherti et al., 2023; Jia et al., 2021; Li et al., 2021; 2022; 2023; Zhu et al., 2024b) trained with language supervision have demonstrated strong capabilities in semantic understanding, particularly in tasks involving open-vocabulary recognition and image-text alignment. However, these models typically lack spatial awareness and fail to capture geometric consistency across views. In contrast, geometric methods (Wang et al., 2024b; Leroy et al., 2024; Wang et al., 2025b) based on multi-view consistency learning focus on spatial correspondence and 3D reconstruction, but typically lack semantic understanding and are difficult to generalize to language-guided tasks. Other works (Ranzinger et al., 2024b;a; 2025; Heinrich et al., 2025; Sarıyıldız et al., 2025) explore multi-teacher feature distillation to combine semantic and geometric knowledge, but their training objectives focus on general-purpose representation fusion rather than geometric awareness.In contrast, we aim to pretrain the vision encoder with both semantic and geometric awareness, tailored for spatial unification.

## 3 METHODOLOGY

### 3.1 PIPELINE OVERVIEW

Fig. 2 presents the overall workflow, with the training pipeline on the left and the inference process on the right. During training, *UniUGG* adopts a three-stage training strategy. First, as shown in Fig. 2 (a), we pretrain the vision encoder to learn geometric-semantic visual representations in stage 1, detailed in Sec. 3.2. Next, as illustrated in Fig. 2 (b), we pretrain the Spatial-VAE in stage 2, which compresses geometric-semantic information into a compact latent space. This module enhances efficient generation and facilitates producing sharper 3D point clouds, as discussed in Sec. 3.3. Finally, we perform unified training for spatial understanding and 3D generation in stage 3, while keeping both the ViT and the VAE encoder frozen, as depicted in Fig. 2 (c). The unified training procedure is elaborated in Sec. 3.3. During inference, *UniUGG* takes images, questions, or view-transformation as input and generates text answers and point clouds, as shown in Fig. 2 (d).

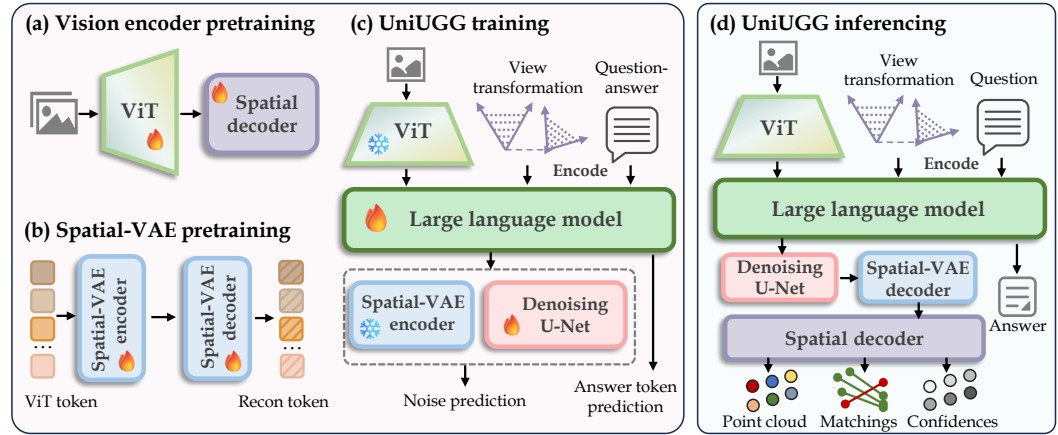

Figure 2: **Pipeline overview of *UniUGG*.** The left illustrates the three-stage training process, and the right shows the inference pipeline for spatial reasoning and 3D generation.

## 3.2 VISION ENCODER PRETRAINING

In this section, we introduce our geometric-semantic learning strategy for vision encoder pretraining, shown in Fig. 3.

**Encoder architecture** Following the design of RADIOv2.5 (Heinrich et al., 2025), we adopt ViT-L/16 (Dosovitskiy et al., 2020) as our basic vision encoder to match the architecture of teachers, which allows us to benefit from its pretraining. Given an input image $\mathcal{I} \in \mathbb{R}^{H \times W \times 3}$, the vision encoder first partitions it into fixed-sized patches of size $p \times p$ pixels, then embeds them into hidden features of dimension $d$ with learnable positional embeddings. After a series of Transformer blocks, the model finally produces a set $Z \in \mathcal{Z}$ of visual representations, where $\mathcal{Z} \in \mathbb{R}^{N \times d}$.

**Multi-view geometric learning** To enhance our encoder's spatial modeling, we adopt the MASt3R (Leroy et al., 2024) framework, retaining its decoder and spatial losses, while replacing the original encoder with ours. As shown in Fig. 3, paired images $\mathcal{I}^i, \mathcal{I}^j$ are encoded by our shared-weight encoder, producing two representations $\mathcal{Z}^i$ and $\mathcal{Z}^j$.

A two-layer visual projector $f_\pi(\cdot)$ with GeLU activation processes each representation independently. Next, the projected features are fed into dual cross-attention decoders, yielding $\mathcal{H}^i, \mathcal{H}^j = \text{Decoder}(f_\pi(\mathcal{Z}^i), f_\pi(\mathcal{Z}^j))$. Finally, pointmaps $X_i^i, X_i^j \in \mathbb{R}^{H \times W \times 3}$ and confidence maps $C^i, C^j$ are regressed from $[\mathcal{H}^i, \mathcal{H}^j]$, along with dense matching descriptors $D^i, D^j \in \mathbb{R}^{H \times W \times d_f}$, via a spatial head.

To extend our encoder for color-awareness, we implement an RGB head to reconstruct color information from encoded representation $\mathcal{Z}$, constrained by a composite loss:

$$\mathcal{L}_{rgb} = \lambda_{L_1} \cdot \|\hat{\mathcal{I}} - \mathcal{I}\|_{L_1} + \lambda_{\text{LP}} \cdot \text{LPIPS}(\hat{\mathcal{I}}, \mathcal{I}), \quad (1)$$

where $\| \cdot \|_{L_1}$ denotes the *L1*-norm, and LPIPS (Zhang et al., 2018) captures perceptual similarity based on deep networks. The final training loss in the spatial branch is defined as:

$$\mathcal{L}_s = \mathcal{L}_{\text{conf}} + \lambda_1 \mathcal{L}_{\text{match}} + \lambda_2 \mathcal{L}_{\text{rgb}}, \quad (2)$$

where $\mathcal{L}_{\text{conf}}$ and $\mathcal{L}_{\text{match}}$ are confidence-aware regression loss and matching loss, respectively, defined in MASt3R.

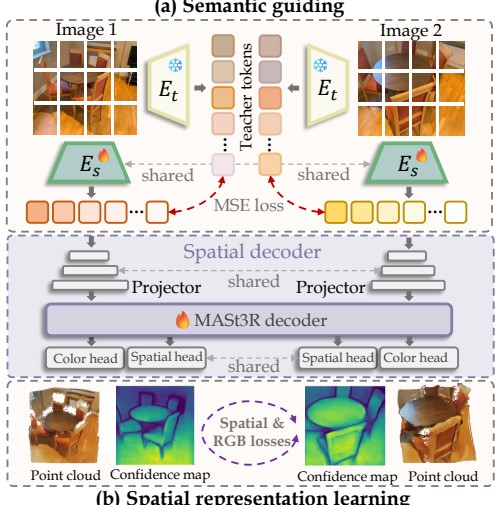

Figure 3: **Overview of our encoder pretraining pipeline.** (a) During semantic guiding, our student encoder learns to mimic the teacher's visual representations. (b) In spatial representation learning, the spatial decoder jointly refines predictions using information from both views.

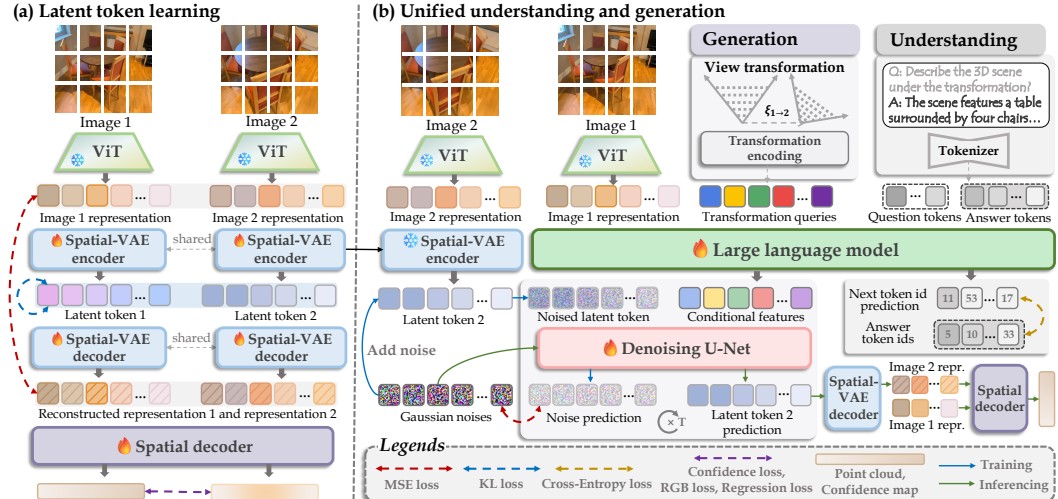

Figure 4: **Overview of *UniUGG* training and inferencing.** (a) In the latent token learning stage (stage 2), visual representation is compressed using the Spatial-VAE, while the spatial decoder is linked for fine-tuning. (b) In the unified learning stage (stage 3), the reference image's visual representation and view transformation are input to an LLM, which outputs conditional features for noise prediction on latent token. The LLM also performs VQA-related training to maintain its understanding capability. During inferencing, *UniUGG* generates the visual representation of the target view, which, together with the reference representation, is decoded into the 3D scene.

Note that in the following, the visual projector, the MASt3R decoder, and the prediction heads are collectively referred to as the spatial decoder, as illustrated in Fig. 3.

**Semantic knowledge guiding** To enhance semantic understanding, we use the pretrained RADIOv2.5 as a teacher to guide our encoder. Given an input image, semantic tokens $\hat{\mathcal{Z}} \in \mathbb{R}^{N \times d}$ are extracted from the teacher and aligned with student tokens $\mathcal{Z}$ via a weighted sum of cosine distance and smooth $L_1$ losses. To improve robustness, the loss is computed over a randomly sampled subset $\mathcal{C}$ of tokens. The guiding loss is defined as:

$$\mathcal{L}_{\text{KD}} = \alpha(1 - \frac{1}{n}\sum_{i \in \mathcal{C}} \cos(z_i, \hat{z}_i)) + \beta\frac{1}{n}\sum_{i \in \mathcal{C}} \|z_i - \hat{z}_i\|_{L_1}, \tag{3}$$

where $n = |\mathcal{C}|$ denotes the number of tokens. This alignment enforces consistency in both feature direction and magnitude.

### 3.3 Unified Understanding and Generation Learning

Leveraging prior knowledge, humans can imagine both geometric and semantic details of unobserved areas from a reference image. With this goal, we aim to enable LLMs to understand and reason scenes through spatial question answering, and imagine novel 3D structures under view changes.

**Overall learning target** Based on our encoder pretraining, we design *UniUGG* for unified spatial understanding and generation, detailed in Fig. 4 (b). For 3D generation, we leverage an LLM in combination with a diffusion model to generate the visual representation of the target view conditioned on a reference image and view transformation. The pretrained spatial decoder then processes the visual representations of both the reference image and the target view, decoding the 3D scene. For 3D understanding, we also perform supervised fine-tuning on the LLM at spatially grounded VQA tasks.

**Latent token learning** Directly generating high-dimensional representations is costly and unstable. To address this, we design and pretrain the Spatial-VAE with an encoder-decoder architecture to compress visual representations into a compact latent space, enabling efficient generation, shown in Fig. 4 (a). Given an image pair $\mathcal{I}^i, \mathcal{I}^j$, our pretrained vision encoder extracts visual representations $\mathcal{Z}^i, \mathcal{Z}^j \in \mathbb{R}^{N_h \times N_w \times d}$, which are encoded into 4-dimensional latent tokens $\mathcal{T}^i, \mathcal{T}^j \in \mathbb{R}^{L_h \times L_w \times 4}$ and then reconstructed back to $\bar{\mathcal{Z}}^i, \bar{\mathcal{Z}}^j$.

The Spatial-VAE optimization is guided by three loss terms: **(i)** Reconstruction loss $\mathcal{L}_{\mathrm{mse}} = \left\| \bar{\mathcal{Z}}^i - \mathcal{Z}^i \right\|^2$. **(ii)** KL loss $\mathcal{L}_{\mathrm{KL}} = D_{\mathrm{KL}}(q(\mathcal{T}^i \mid \mathcal{Z}^i) \parallel p(\mathcal{T}^i)) + D_{\mathrm{KL}}(q(\mathcal{T}^j \mid \mathcal{Z}^j) \parallel p(\mathcal{T}^j))$, where $D_{\mathrm{KL}}$ denotes the Kullback-Leibler divergence for latent distribution regularization. **(iii)** Spatial loss $\mathcal{L}_{\mathrm{s}}$, defined in Eq. 2. Due to discrepancies between reconstructed and original representations, the pretrained spatial decoder may struggle to deal with the reconstructed representation. To address this and further guide compression, we feed the reconstructed representations into the spatial decoder and fine-tune it jointly with the Spatial-VAE in an end-to-end manner. The overall latent token learning loss $\mathcal{L}_{\mathrm{vae}} = \mathcal{L}_{\mathrm{s}} + \mathcal{L}_{\mathrm{mse}} + \gamma \mathcal{L}_{\mathrm{KL}}$, where $\gamma$ is the weight for the KL loss term.

**Spatial generation learning** As shown in Fig. 4 (b), with the pretrained Spatial-VAE, the 3D generation can be modeled as generating the latent token conditioned on a reference image and a view transformation. This latent token is then decoded into the visual representation of the target-view using the VAE decoder. Subsequently, the 3D scene is decoded by the fine-tuned spatial decoder to the visual representations from both the reference and target views. Therefore, spatial generation learning is naturally the process of conditional noise prediction on the noisy latent token.

During training, the relative view transformation between $\mathcal{I}^i$ and $\mathcal{I}^j$ is encoded as a Plücker raymap (Plucker, 1865), represented as $\mathbf{P} \in \mathbb{R}^{N_h \times N_w \times 6}$. This raymap is then transformed into queries $\mathbf{q}$ using an MLP, so that suitable for processing by LLM. Subsequently, we feed the visual representation of $\mathcal{I}^i$, $\mathcal{Z}^i$, along with the transformation queries $\mathbf{q}$, into the LLM, which generates the conditional features $\mathbf{C}$.

The next step involves training the model to predict the noise in the noisy latent tokens. We encode $\mathcal{I}^j$'s visual representation $\mathcal{Z}^j$ into the latent token $\mathcal{T}^j$ via the pretrained VAE encoder. Gaussian noise is progressively added to the latent token $\mathcal{T}^j$ over several timesteps, creating a noisy latent token $\tilde{\mathcal{T}}_t^j$ for each timestep $t$. Specifically, the noise is added according to a schedule that increases with each timestep. The noisy latent token at each timestep is then passed through the denoising diffusion model along with the corresponding conditional features $\mathbf{C}$. At each timestep $t$, the model learns to predict the noise $\epsilon_\theta(\tilde{\mathcal{T}}_t^j | \mathbf{C}, t)$ added to the noisy latent token. The training target is minimizing the discrepancy between the predicted noise $\epsilon_\theta$ and the ground-truth noise $\epsilon$:

$$\mathcal{L}_{\mathrm{gen}} = \mathbb{E}_{\mathcal{T}^j, \epsilon \sim \mathcal{N}(0,1), t} \left[ \left\| \epsilon_\theta(\tilde{\mathcal{T}}_t^j | \mathbf{C}, t) - \epsilon \right\|^2 \right]. \tag{4}$$

At inference, we start from a noisy latent token $\tilde{\mathcal{T}}_T \sim \mathcal{N}(0, 1)$ and iteratively denoise it using the reverse diffusion process. At each timestep $t$, the model predicts the noise to be removed, updating the latent token by $\tilde{\mathcal{T}}_{t-1} = \tilde{\mathcal{T}}_t - \epsilon_\theta(\tilde{\mathcal{T}}_t | \mathbf{C}, t)$. Here, $\mathbf{C}$ represents the conditional feature generated by the LLM, which takes the reference visual representation $\mathcal{Z}_r$ and an arbitrary view transformation as input. After $T$ steps, the final latent token $\tilde{\mathcal{T}}_0$ is decoded into target view's visual representation $\mathcal{Z}_v$ by the Spatial-VAE decoder, and the full 3D structure is then decoded by the fine-tuned spatial decoder using both $\mathcal{Z}_r$ and $\mathcal{Z}_v$.

**Spatial understanding learning** Given an input image $\mathcal{I}$ and a question $\mathcal{Q}$, the model predicts an answer sequence $\mathbf{a} = \{a_0, a_1, ..., a_N\}$ in an autoregressive manner. Firstly, the question $\mathcal{Q}$ is tokenized into language embeddings $\mathbf{q} = \{q_0, q_1, ..., q_m\}$. Together with the image representation $\mathcal{Z}$, we feed these language tokens into the LLM. At step $t$, the LLM produces a distribution over the next token conditioned on ground-truth prefix $a_{<t}$. *UniUGG* is trained with teacher forcing using a token-level cross-entropy loss $\mathcal{L}_{\mathrm{vqa}} = -\sum_{t=1}^{N} \log p_\theta(a_t | \mathcal{Z}, \mathbf{q}, a_{<t})$. Here, $a_t$ is the ground-truth token at $t$. At inference, the prefix $a_{<t}$ is replaced by the previously generated tokens $\hat{a}_{<t}$.

# 4 EXPERIMENTS

## 4.1 IMPLEMENTATION DETAILS

**Vision encoder pretraining** We initialize our vision encoder with RADIOv2.5-L (Heinrich et al., 2025), a ViT-Large model with 24 Transformer layers and a hidden size of 1024. The encoder is followed by a ViT-Base decoder initialized from MASt3R (Leroy et al., 2024). We pretrained our encoder on a mixture of ARKitScenes (Baruch et al., 2021) and ScanNet++(Yeshwanth et al., 2023) to capture geometric capabilities, and LAION-400M(Schuhmann et al., 2021) to capture semantic diversity. More training details can be found in Appendix A.2.

**Training *UniUGG*** The Spatial-VAE is trained on 2M co-viewing pairs from ARKitScenes and ScanNet++, with the KL loss weight $\gamma = 0.0001$. The detailed Spatial-VAE architecture can be found in Appendix A.2. For unified training in stage 3, the vision encoder and the Spatial-VAE are frozen, with only the LLM, projector, and diffusion model optimized. The training process follows three steps: **(i)** Projector is trained on LCS-558K (Liu et al., 2023) to align patch-level features with the LLM embedding space; **(ii)** LLM, diffusion model, and projector are jointly optimized on 2.4M spatial instruction-following samples from ShareGPT4V (Chen et al., 2024d) and ALLaVA (Chen et al., 2024b), along with 2M co-viewing pairs from ARKitScenes and ScanNet++; **(iii)** Model is further finetuned on SPAR (Zhang et al., 2025a), EMOVA (Chen et al., 2024c), and an additional 2M spatial sample pairs to enhance generalization in spatial QA and 3D generation tasks.

We use Qwen2.5-3B-Instruct (Bai et al., 2023) as the LLM backbone and stable-diffusion-v1-5 (Rombach et al., 2022) as the diffusion model. The AdamW optimizer with cosine learning rate decay and a warm-up ratio of 0.03 is used. The learning rate is set to $1 \times 10^{-3}$ for step (i) and $2 \times 10^{-5}$ for steps (ii) and (iii) in unified training. The global batch size is set to 256. Additional 0.25M samples are used for generation comparison, separate from the training data.

**Training cost and computational resources** Our pipeline adopts a three-stage training strategy as illustrated in Fig. 2. In stage 1, we pretrain the geometric-semantic encoder for 25 hours using 8× NVIDIA A6000 GPUs. In stage 2, the Spatial-VAE module is pretrained for 12 hours on the same 8× A6000 GPUs. In stage 3, we train the full *UniUGG* model on both spatial understanding and 3D generation tasks for about 46 hours on a cluster with 8 nodes, each equipped with 8 Ascend NPUs. During inference, we run our model on a single NVIDIA A6000 GPU. For spatial reasoning, given 16 input images at a resolution of 224×224, *UniUGG* achieves an inference latency of 350ms, utilizing bf16 precision and FlashAttention for acceleration. For 3D generation, *UniUGG* takes approximately 1.2s to generate a point cloud from a single reference image.

| | Casual | | | | | | | | | MipNeRF360 | | | | | | | | |
|---|---|---|---|---|---|---|---|---|---|---|---|---|---|---|---|---|---|---|
| | Geometry | | | Texture | | | All | | | Geometry | | | Texture | | | All | | |
| Feature | PSNR↑ | SSIM↑ | LPIPS↓ | PSNR↑ | SSIM↑ | LPIPS↓ | PSNR↑ | SSIM↑ | LPIPS↓ | PSNR↑ | SSIM↑ | LPIPS↓ | PSNR↑ | SSIM↑ | LPIPS↓ | PSNR↑ | SSIM↑ | LPIPS↓ |
| DINOv2 | 19.42 | .6524 | .3698 | 17.64 | .5701 | .3754 | 19.21 | .6535 | .4023 | 20.81 | .4946 | .3953 | 19.05 | .4495 | .3821 | 20.75 | .4924 | **.4684** |
| CLIP | 19.21 | .6552 | .3719 | 17.46 | .5669 | .3743 | 19.05 | .6582 | .4084 | 20.80 | .4982 | .3913 | 19.28 | .4543 | .3807 | 20.88 | .4984 | .4773 |
| DUSt3R | 19.29 | .6562 | .3580 | 17.54 | .5693 | .3750 | 19.19 | .6556 | .4050 | 20.82 | .5008 | .3795 | 19.10 | .4489 | .3816 | 21.02 | .5048 | .4752 |
| VGGT | 19.36 | .6590 | .3549 | 17.47 | .5645 | .3751 | 19.23 | .6604 | .4103 | 20.93 | .5120 | .3639 | 19.25 | .4497 | .3828 | 21.17 | .5102 | .4892 |
| RADIO | 19.54 | .6545 | .3465 | 17.52 | .5666 | .3748 | 18.67 | .6533 | .4216 | 20.87 | .5100 | .3620 | 19.35 | .4550 | .3819 | 20.91 | .5067 | .5127 |
| MASt3R | 19.30 | .6550 | .3576 | 17.59 | .5708 | .3722 | **19.37** | .6588 | .4027 | 20.92 | .5093 | .3745 | 19.21 | .4540 | .3803 | 20.92 | .5054 | .4749 |
| Ours | **19.80** | **.6643** | **.3449** | **17.81** | **.5850** | **.3559** | 19.18 | **.6693** | **.3955** | **21.28** | **.5337** | **.3562** | **19.72** | **.4848** | **.3595** | **21.31** | **.5264** | .4698 |

Table 1: **Results of novel view synthesis metrics on Feat2GS benchmark.** Our encoder outperforms others in most datasets across Geometry, Texture, and All probing modes. The best results are marked in **bold**, and the second best in underlined.

## 4.2 EVALUATION OF THE GEOMETRIC-SEMANTIC ENCODER

**Evaluation on Feat2GS benchmark** The Feat2GS benchmark (Chen et al., 2024g) evaluates novel view synthesis as a proxy task for assessing 3D awareness, which defines three evaluation modes: **(i)** Geometry: only geometry parameters are predicted from encoder features, while the texture is free-optimized for novel view rendering; **(ii)** Texture: only texture is predicted, with the geometry free-optimized; **(iii)** All: both geometry and texture are predicted from features. The encoders compared include DINOv2 (Oquab et al., 2023), CLIP (Radford et al., 2021), DUST3R encoder (Wang et al., 2024b), VGGT encoder (Wang et al., 2025a), AM-RADIO (Ranzinger et al., 2024b), and MASt3R encoder. As shown in Tab. 1, our vision encoder outperforms baselines in all three probing modes, achieving significant improvements. Detailed results are provided in Appendix A.3.1.

**Evaluation on semantic perception and 3D vision tasks** We comprehensively evaluate our pretrained vision encoder across a diverse set of tasks, including monocular and video depth estimation, image-level reasoning, and pixel-level visual understanding. Our method achieves highly competitive performance across all evaluations. Detailed results are provided in Appendix A.3.1.

**Downstream task performance** We assess the spatial understanding performance of our pretrained encoder by integrating it into a unified Vision-Language Model architecture based on Qwen2.5-3B-Instruct (Bai et al., 2023). We evaluate on a wide range of vision-language reasoning benchmarks.

| Method | Para. (M) | VSI | BLINK | SPAR | | | Seed[1] | Real World |
|---|---|---|---|---|---|---|---|---|
| | | | | Low | Med. | High | | |
| Qwen2.5-ViT | 669 | 35.56 | 37.81 | 36.50 | 36.67 | 39.89 | 41.81 | 44.97 |
| CLIP-L/14 | 305 | _40.08_ | 40.45 | 44.13 | 43.67 | **52.33** | 69.14 | 54.38 |
| SigLIP-L/16 | 316 | 23.81 | 39.08 | 41.75 | 34.67 | 43.11 | 56.31 | 45.23 |
| MASt3R Enc. | 303 | 39.14 | 40.93 | 50.00 | 42.33 | 48.22 | 56.96 | 50.07 |
| RADIOv2.5-L | 320 | 39.75 | _42.92_ | _50.44_ | _47.95_ | _52.13_ | **72.09** | _57.38_ |
| _UniUGG_ Enc. | 320 | **42.18** | **44.40** | **50.82** | **49.07** | 51.89 | _71.65_ | **58.56** |

Table 2: **Comparison of encoder performance on downstream vision-language reasoning benchmarks.** The VLM architecture is based on Qwen2.5-3B-Instruct, and all encoders are trained under the same settings to ensure fairness.

Spatial and geometric abilities are assessed through VSI-Bench (Yang et al., 2025), SPAR (Zhang et al., 2025a) and BLINK (Fu et al., 2024b), while general language understanding is tested on RealWorldQA (Miyanishi et al., 2021) and SEED-I (Li et al., 2024b). Compared models include both semantic-oriented encoders (CLIP-L/14, SigLIP-L/16 (Zhai et al., 2023), RADIOv2.5-L) and geometry-aware design (MASt3R encoder). All encoders are initialized from pretrained checkpoints and fine-tuned with the LLM under the same settings for fair comparison.

As shown in Tab. 2, on VSI-Bench, BLINK, and SPAR, our encoder (_UniUGG_ Enc.) demonstrates clear advantages on spatial reasoning tasks, which require spatial relational understanding and geometric abstraction. Moreover, our method also achieves competitive performance on general QA benchmarks like RealWorldQA and SEED-I, showing that spatial enhancement does not significantly impair semantic generalization. Compared to the geometry-focused MASt3R encoder, our encoder shows more consistent performance across modalities. In general, our encoder can bridge geometry and semantics in a unified representation, balancing spatial perception and high-level semantics.

### 4.3 EVALUATION OF SPATIAL UNDERSTANDING

To evaluate _UniUGG_'s spatial reasoning ability, we assess it on representative benchmarks, i.e., VSI-Bench, BLINK, 3DSRBench (Ma et al., 2024), and SPAR. We use three open-source LMMs—LLaVA (Liu et al., 2024b), InternVL2.5 (Chen et al., 2024h), Qwen2.5VL (Bai et al., 2025)—and one proprietary LMM, GPT-4o (Achiam et al., 2023), along with the state-of-the-art 2D unified framework Janus-Pro (Chen et al., 2025) for comparison. Results shown in Tab. 3, our _UniUGG_ achieves superior performance across most benchmarks. In particular, on VSI-Bench, our model outperforms the second-best one by 17.9%. It demonstrates that _UniUGG_ can capture fine-grained spatial relations by jointly modeling 3D structure and visual-language reasoning. Additional evaluations on SQA3D (Ma et al., 2022), ScanQA (Azuma et al., 2022), and ScanRefer (Chen et al., 2020) are provided in Appendix A.3.2. It should be noted that _UniUGG_ is designed for multi-view spatial understanding, where the model learns geometry from 2D inputs. Therefore, on these 3D benchmarks, our model still shows a performance gap compared to 3D-enhanced methods.

### 4.4 EVALUATION OF 3D GENERATION

**Quantitative generation comparison and ablations** We compare _UniUGG_ with baselines and perform ablation studies to evaluate the quality of the generated outputs. Given a reference image and a view transformation, the setup generates the corresponding spatial structure for the novel view. The generated point cloud is then projected back onto the image plane, producing a colored 2D image. These generated images are compared to the real images (Ground truth) using the Fréchet inception distance (FID), kernel inception distance (KID), and LPIPS. Quantitative and qualitative results are presented in Tab. 3 and Fig. 5, respectively.

The encoder from our pretraining strategy (ID g) significantly improves generation quality, outperforming both the RADIOv2.5-L (ID a) and MASt3R encoder (ID b). This shows that simply incorporating geometric or semantic information is insufficient, and fusing both is more effective in a unified framework. From the _UniUGG_ settings, we observe a notable performance drop when the spatial decoder is not fine-tuned during VAE training (ID c). Additionally, omitting the Spatial-VAE and diffusion models (ID d), and having the LLM directly predict the target-view representation, results in suboptimal performance. By the way, removing the Spatial-VAE only and training generation

| Method | VSI | BLINK | 3DSR | SPAR Low | SPAR Med. | SPAR High | SPAR Avg. |
|---|---|---|---|---|---|---|---|
| LLaVA-v1.5-7B | 18.0 | 37.1 | 38.1 | 10.9 | 26.5 | 34.1 | 23.7 |
| LLaVA-NeXT-7B | 20.6 | 41.8 | 48.4 | 8.5 | 4.8 | 20.2 | 13.2 |
| InternVL2.5-8B | 32.5 | 54.8 | 50.9 | 29.5 | 31.9 | 43.8 | 36.3 |
| Qwen2.5-VL-7B | 30.3 | 56.4 | 48.4 | 28.8 | 23.0 | 40.3 | 33.1 |
| GPT-4o | 34.0 | 60.0 | 44.2 | 36.9 | 26.5 | 43.8 | 38.1 |
| ∗Janus-Pro-1B | - | 38.9 | 50.0 | 10.7 | 24.7 | 30.8 | 20.6 |
| ∗Janus-Pro-7B | - | 40.5 | 53.7 | 27.3 | 24.6 | 33.9 | 28.6 |
| *UniUGG*-3B (Ours) | 40.1 | 43.6 | 52.1 | 50.8 | 41.7 | 45.7 | 47.2 |

| ID Method | ARKitScenes FID↓ | KID↓ | LPIPS↓ | ScanNet++ FID↓ | KID↓ | LPIPS↓ |
|---|---|---|---|---|---|---|
| (a) w/ RADIO | 64.16 | .0518 | .4904 | 73.69 | .0614 | .4629 |
| (b) w/ MASt3R Enc. | 81.18 | .0691 | .5076 | 86.79 | .0803 | .5242 |
| (c) w/o Dec. finetune | 149.97 | .1447 | .5301 | 168.05 | .1686 | .4945 |
| (d) w/o Diffusion | 87.51 | .0672 | .4494 | 114.93 | .0955 | .4345 |
| (e) CUT3R | 138.54 | .1128 | .5758 | 130.76 | .1051 | .5637 |
| (f) LVSM | 269.45 | .3088 | .5067 | 414.63 | .5117 | .5865 |
| (g) *UniUGG* (Ours) | 55.01 | .0425 | .4849 | 55.64 | .0442 | .4263 |

Table 3: **Comparison of 3D understanding and generation performance.** *Left*: 3D understanding performance on various spatial reasoning benchmarks. ∗denotes 2D understanding and generation method. *Right*: Quantitative spatial generation comparison on ARKitScenes and ScanNet++ datasets. ID(a) to ID(d) represent the ablation of our model.

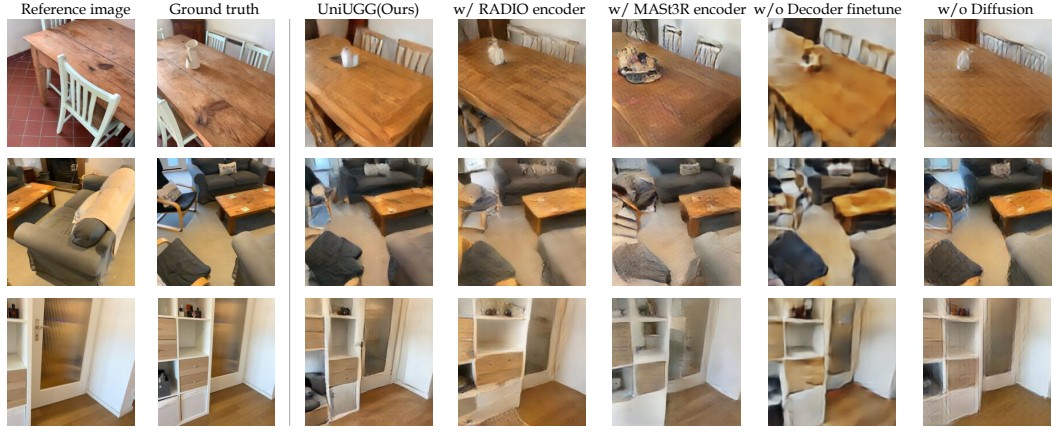

Reference image  Ground truth  UniUGG(Ours)  w/ RADIO encoder  w/ MASt3R encoder  w/o Decoder finetune  w/o Diffusion

Figure 5: **Qualitative ablation on 2D projected views from 3D generation.** Our *UniUGG*, including the geometric-semantic encoder, Spatial-VAE, and associated training paradigm, leads to noticeably better generation results, in terms of geometric accuracy and color consistency.

directly on the original representation also fails to generate valid results. These results demonstrate the Spatial-VAE and related training paradigms are the key to successful 3D generation. Finally, we compare *UniUGG* with baselines, CUT3R (Wang et al., 2025b) and LVSM (Jin et al., 2025). While CUT3R (ID e) predicts 3D structures from pre-observed data and raymap, and LVSM (ID f) generates target-view 2D images, both fall short in performance due to their lack of imaginative capabilities. This highlights the superiority of our method in 3D generation.

**Qualitative understanding and generation comparison** We further qualitatively assess our *UniUGG* with CUT3R, shown in Fig. 6. From the perspective of the generated area, *UniUGG* accurately identifies which parts of the geometric structure need to be generated. Additionally, in terms of texture and semantic details, *UniUGG* effectively leverages the reference image to plausibly infer new structures, such as windows and chairs. We also demonstrate the understanding capabilities of *UniUGG* by generating captions for the scene from the generated visual representations. *UniUGG* can provide accurate descriptions of the 3D structure, even for parts that were previously unseen. In contrast, the baseline struggles to complete missing regions and lacks coherence in structural details, let alone understanding the scene. These results highlight the strengths of *UniUGG* in unified spatial understanding and 3D generation. We provide additional visualizations—including feature matching, evaluation under extreme view transformations, and failure cases—in Appendix A.3.3.

## 5 CONCLUSION AND LIMITATIONS

In this paper, we introduce *UniUGG*, the first unified framework for spatial generation and understanding, capable of spatial-level VQA and generating 3D scenes. We propose a geometric-semantic learning strategy to pretrain the vision encoder, enhancing its spatial modeling capabilities. This

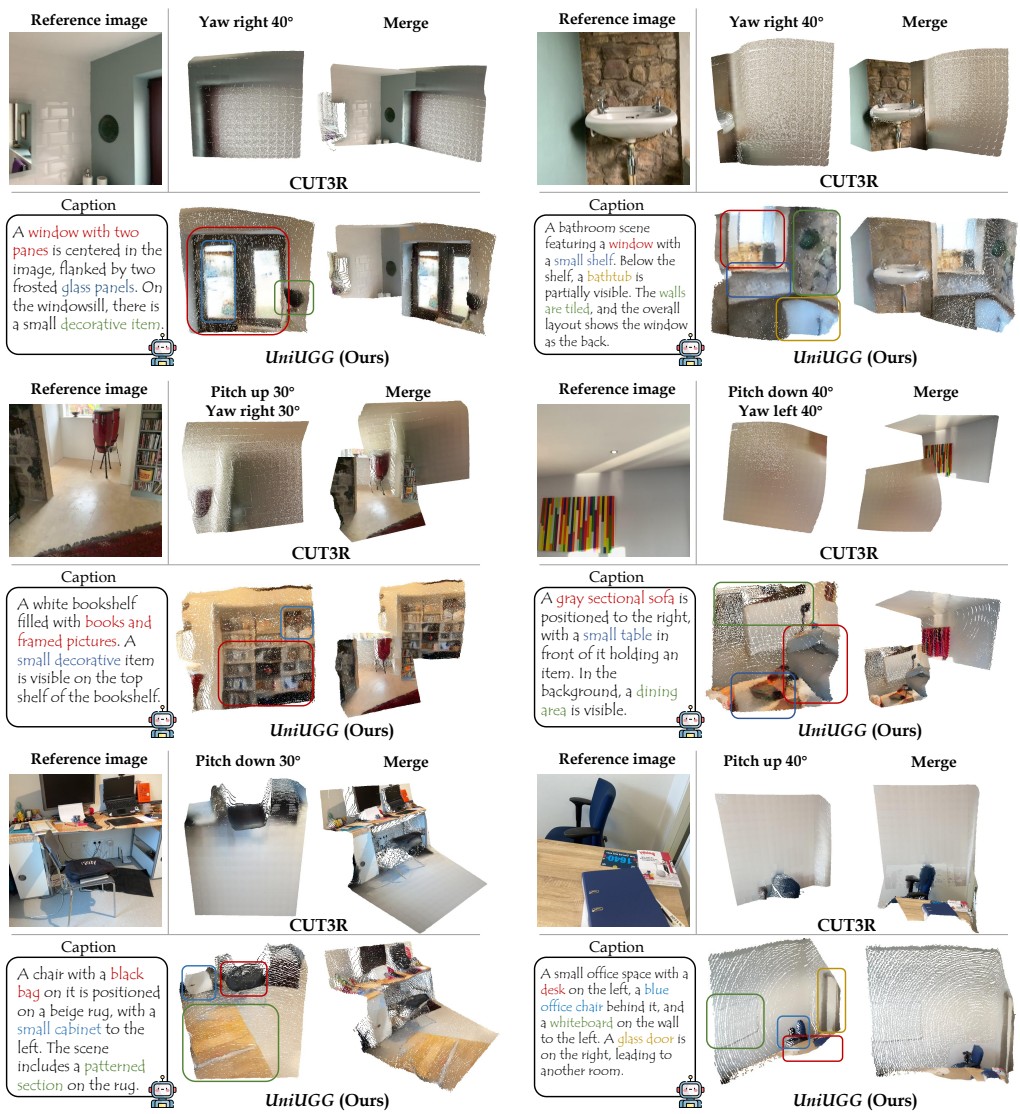

Figure 6: **Qualitative 3D generation comparison.** *UniUGG* accurately captures the input view transformation and leverages the reference image to 'imagine' fine-grained spatial structures, and outputs correct captioning. In contrast, the baseline method only produces coarse and fuzzy geometry.

significantly improves both the generation and understanding aspects of our unified framework and yields strong performance on downstream tasks. Moreover, we design the Spatial-VAE for achieving efficient 3D generation, and link the spatial decoder for fine-tuning to ensure sharper 3D scene decoding. Extensive evaluations showcasing *UniUGG*'s ability to handle both 3D generation and spatial VQA tasks effectively. However, further enhancing 3D generation capabilities in unified models remains a key challenge for future research. Our framework still has several limitations, including the lack of controllable generation driven by language and the inability to perform freeform editing of the generated content. As a preliminary exploration toward unified 3D modeling, our method does not yet support interactive multi-round scene generation and editing, which also needs to be solved in the future.

## 6 ACKNOWLEDGMENTS

This work was supported in part by New Generation Artificial Intelligence-National Science and Technology Major Project (2025ZD0123004), Ningbo grant (2025Z038) and National Natural Science Foundation of China (Grant No. 62376060).

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

# A APPENDIX

In Appendix, we provide additional technical details and more detailed experimental validations in terms of our method, comparison, and visualization.

## A.1 USE OF LARGE LANGUAGE MODELS

We used GPT-4o to polish the language and improve the clarity of our manuscript, such as improving grammar and phrasing. The model did not contribute to the core research content of the manuscript, including mathematical formulations, algorithms, and related results.

## A.2 ADDITIONAL IMPLEMENTATION DETAILS

**Vision encoder pretraining** In the experiments, the patch size is set to $16 \times 16$, producing 768 tokens of dimension 1024. The training uses AdamW optimizer with cosine decay over 5 epochs, with loss weights $\lambda_{L_1} = 1.0$, $\lambda_{LP} = 0.5$, $\lambda_1 = 1.0$, $\lambda_2 = 1.0$, and semantic guiding loss weights $\alpha = 0.9$, $\beta = 0.1$.

**Downstream task evaluation** To assess the effectiveness of the pretrained encoder across downstream tasks, we adopt an experimental configuration in which the encoder, projector, and LLM are jointly trained in an end-to-end fashion. While this setting follows the stage 3 training pipeline as described in the *Sec. 4.1 'Training UniUGG' part of the main text*, it differs in two key aspects: the encoder is updated during training, and no additional co-viewing sample pairs are included. This design allows us to evaluate the performance of different encoders on both spatial understanding and general reasoning tasks fairly.

We assess the downstream performance of our pretrained encoder and other encoders by integrating them into a unified Vision-Language Model (VLM) architecture based on Qwen2.5-3B-Instruct (Bai et al., 2023). All models are trained with the same pipeline, jointly optimizing the encoder, visual projector, and LLM. This design ensures fair comparison under identical supervision and model capacity. All encoders are initialized from their respective pretrained checkpoints and jointly finetuned with the LLM under the same settings to ensure fair comparison.

**Spatial-VAE architecture.** The detailed Spatial-VAE architecture is provided in Tab. 4. The architecture follows an encoder-decoder design tailored for compressing and reconstructing visual representations. The encoder first reshapes the visual representation inputs into a 2D feature map, applies a series of convolutional and attention layers, and finally outputs the latent mean and variance for sampling. The decoder mirrors this process by reconstructing the visual representations from the

| Layer | Description | Output shape |
|---|---|---|
| | **Sptatial-VAE encoder** | |
| 0 | Reshaped input | [b, 1024, 14, 14] |
| 1 | Initial convolution | [b, 256, 14, 14] |
| 2 | Upsample 1 (convtranspose) | [b, 128, 28, 28] |
| 3 | Upsample 2 (convolution) | [b, 128, 28, 28] |
| 4 | Transformer blocks | [b, 784, 128] |
| 5 | Flattened attention output | [b, 128, 28, 28] |
| 6 | $\mu$ convolution | [b, 4, 28, 28] |
| 7 | Log-variance convolution | [b, 4, 28, 28] |
| 8 | Reparameterization | [b, 4, 28, 28] |
| 9 | KL divergence loss | scalar (mean of log variance) |
| | **Sptatial-VAE decoder** | |
| 0 | Input | [b, 4, 28, 28] |
| 1 | Pre-convolution | [b, 128, 28, 28] |
| 2 | Flattened attention output | [b, 784, 128] |
| 3 | Transformer blocks | [b, 784, 128] |
| 4 | Reshaped attention output | [b, 128, 28, 28] |
| 5 | Downsample 1 (convolution) | [b, 128, 28, 28] |
| 6 | Downsample 2 (convolution) | [b, 256, 14, 14] |
| 7 | Final convolution | [b, 1024, 14, 14] |
| 8 | Output reshaped | [b, 196, 1024] |

Table 4: **Detailed Spatial-VAE architecture.** Our model follows an encoder-decoder design.

sampled latent features using a combination of attention blocks and convolutional layers. Transformer-based attention modules are used in both the encoder and decoder to model long-range dependencies across spatial positions, enhancing semantic fidelity during compression and reconstruction.

## A.3 MORE EXPERIMENTAL RESULTS

### A.3.1 EVALUATION OF THE GEOMETRIC-SEMANTIC ENCODER

**Evaluation on Feat2GS benchmark** We provide comprehensive and detailed results on Feat2GS benchmark (Chen et al., 2024g), as shown in Tab. 5. Results indicate that our encoder leads to the best performance on most datasets in Geometry, Texture, and All probing modes.

**Single-frame and video depth estimation** Following MonST3R (Zhang et al., 2024), we evaluate single-frame depth on the NYU-v2 (Silberman et al., 2012) dataset and video depth on the BONN (Silberman et al., 2012) dataset, which cover dynamic and static scenes. These datasets are excluded from training, enabling zero-shot performance evaluation across domains. Our evaluation metrics include absolute relative error (Abs Rel) and percentage of predicted depths within a 1.25-factor of true depth ($\delta < 1.25$). Following (Wang et al., 2024b), single-frame evaluation adopts per-frame median scaling, and video evaluation aligns a single scale and/or shift factor per sequence.

We compared our methods with DUSt3R (Wang et al., 2024b), MASt3R (Leroy et al., 2024), Spann3R (Wang & Agapito, 2024), and MonST3R, where these baselines are specially designed for 3D tasks. As shown in Tab. 6 (left), our method achieves competitive results compared to baselines, and even outperforms MASt3R in both single-frame and video depth evaluation.

**Image/pixel level evaluation** To assess the performance of our encoders, we adopt a set of representative metrics following (Ranzinger et al., 2024b). For image-level reasoning, we evaluate our encoder using Top-1 k-NN accuracy and zero-shot accuracy on the ImageNet-1K dataset (Deng et al., 2009). The zero-shot accuracy is computed using the CLIP language model (Radford et al., 2021). For the k-NN evaluation, we first extract the summary feature for all training images. Then, for each validation image, we identify the $k$ nearest neighbors in the feature space and predict the label based on a weighted vote of these neighbors. We also evaluate the generalization performance of our encoder on pixel-level visual tasks, including segmentation mIOU on ADE20k (Zhou et al., 2019) and PascalVOC2012 (Everingham et al., 2015) datasets.

The encoders compared include SAM-H/16 (Kirillov et al., 2023), OpenAI CLIP-L/14 (Radford et al., 2021), SigLIP-L/14 (Zhai et al., 2023) , DINOv2-g/14-reg (Darcet et al., 2023), DUSt3R

| Feature | LLFF Geometry | | | Texture | | | All | | | DL3DV Geometry | | | Texture | | | All | | | Casual Geometry | | | Texture | | | All | | |
|---|---|---|---|---|---|---|---|---|---|---|---|---|---|---|---|---|---|---|---|---|---|---|---|---|---|---|---|
| | PSNR↑ | SSIM↑ | LPIPS↓ | PSNR↑ | SSIM↑ | LPIPS↓ | PSNR↑ | SSIM↑ | LPIPS↓ | PSNR↑ | SSIM↑ | LPIPS↓ | PSNR↑ | SSIM↑ | LPIPS↓ | PSNR↑ | SSIM↑ | LPIPS↓ | PSNR↑ | SSIM↑ | LPIPS↓ | PSNR↑ | SSIM↑ | LPIPS↓ | PSNR↑ | SSIM↑ | LPIPS↓ |
| DINOv2 | 19.77 | .7345 | .2226 | 19.04 | .7133 | .2254 | 19.91 | .7163 | .2637 | 19.47 | .7293 | .3288 | 18.00 | .6805 | .3223 | 19.27 | .7317 | .3479 | 19.42 | .6524 | .3698 | 17.64 | .5701 | .3754 | 19.21 | .6535 | .4023 |
| CLIP | 19.78 | .7378 | .2221 | 19.02 | .7113 | .2276 | 19.74 | .7136 | .2822 | 19.53 | .7295 | .3304 | 18.05 | .6771 | .3235 | 19.22 | .7310 | .3563 | 19.21 | .6552 | .3719 | 17.46 | .5669 | .3743 | 19.05 | .6582 | .4084 |
| DUSt3R | 19.88 | .7442 | .2123 | 19.01 | .7120 | .2262 | 19.87 | .7190 | .2691 | 19.64 | .7338 | .3196 | 18.01 | .6815 | .3219 | 19.39 | .7360 | .3458 | 19.29 | .6562 | .3580 | 17.54 | .5693 | .3750 | 19.19 | .6556 | .4050 |
| VGGT | 19.85 | .7450 | .2127 | 19.05 | .7120 | .2273 | 19.86 | .7165 | .2911 | 19.65 | .7372 | .3143 | 18.05 | .6770 | .3237 | 19.38 | .7358 | .3534 | 19.36 | .6590 | .3549 | 17.47 | .5645 | .3751 | 19.23 | .6604 | .4103 |
| RADIO | 19.73 | .7402 | .2207 | 19.06 | .7101 | .2301 | 19.56 | .6999 | .3252 | 19.48 | .7313 | .3139 | 18.03 | .6748 | .3254 | 19.20 | .7316 | .3654 | 19.54 | .6545 | .3465 | 17.52 | .5666 | .3748 | 18.67 | .6533 | .4216 |
| MASt3R | 19.89 | .7447 | .2123 | 19.01 | .7115 | .2261 | 19.99 | .7250 | .2657 | 19.64 | .7334 | .3188 | 18.07 | .6813 | .3211 | 19.41 | .7373 | .3464 | 19.30 | .6550 | .3576 | 17.59 | .5708 | .3722 | 19.37 | .6588 | .4027 |
| Ours | 19.52 | .7457 | .2073 | 18.79 | .7140 | .2201 | 19.71 | .7199 | .2785 | 18.32 | .7085 | .3382 | 17.29 | .6626 | .3350 | 18.15 | .7147 | .3603 | 19.80 | .6643 | .3449 | 17.81 | .5850 | .3559 | 19.18 | .6693 | .3955 |

| Feature | MipNeRF360 Geometry | | | Texture | | | All | | | MVImgNet Geometry | | | Texture | | | All | | | Tanks and Temples Geometry | | | Texture | | | All | | |
|---|---|---|---|---|---|---|---|---|---|---|---|---|---|---|---|---|---|---|---|---|---|---|---|---|---|---|---|
| | PSNR↑ | SSIM↑ | LPIPS↓ | PSNR↑ | SSIM↑ | LPIPS↓ | PSNR↑ | SSIM↑ | LPIPS↓ | PSNR↑ | SSIM↑ | LPIPS↓ | PSNR↑ | SSIM↑ | LPIPS↓ | PSNR↑ | SSIM↑ | LPIPS↓ | PSNR↑ | SSIM↑ | LPIPS↓ | PSNR↑ | SSIM↑ | LPIPS↓ | PSNR↑ | SSIM↑ | LPIPS↓ |
| DINOv2 | 20.81 | .4946 | .3953 | 19.05 | .4495 | .3821 | 20.75 | .4924 | .4684 | 19.35 | .5896 | .3246 | 16.88 | .5359 | .3344 | 19.43 | .5943 | .3674 | 18.71 | .6432 | .3772 | 17.58 | .6214 | .3348 | 18.43 | .6443 | .4064 |
| CLIP | 20.80 | .4982 | .3913 | 19.28 | .4543 | .3807 | 20.88 | .4984 | .4773 | 19.41 | .5945 | .3098 | 16.96 | .5362 | .3358 | 19.37 | .5969 | .3695 | 18.92 | .6463 | .3729 | 17.81 | .6226 | .3316 | 18.75 | .6515 | .4052 |
| DUSt3R | 20.82 | .5008 | .3795 | 19.10 | .4489 | .3816 | 21.02 | .5048 | .4752 | 19.47 | .6004 | .3073 | 16.88 | .5348 | .3334 | 19.43 | .5937 | .3674 | 18.85 | .6458 | .3715 | 17.53 | .6222 | .3328 | 18.61 | .6477 | .4023 |
| VGGT | 20.93 | .5120 | .3639 | 19.25 | .4497 | .3828 | 21.17 | .5102 | .4892 | 19.48 | .6019 | .2975 | 17.00 | .5373 | .3346 | 19.58 | .5987 | .3748 | 19.21 | .6615 | .3547 | 17.75 | .6221 | .3319 | 19.04 | .6593 | .4017 |
| RADIO | 20.87 | .5100 | .3620 | 19.35 | .4550 | .3819 | 20.91 | .5067 | .5127 | 19.54 | .6105 | .2949 | 16.99 | .5373 | .3366 | 19.60 | .5955 | .3946 | 19.19 | .6612 | .3480 | 17.84 | .6225 | .3321 | 19.01 | .6574 | .4109 |
| MASt3R | 20.92 | .5093 | .3745 | 19.21 | .4540 | .3803 | 20.92 | .5054 | .4749 | 19.49 | .6008 | .3032 | 16.91 | .5350 | .3337 | 19.49 | .5983 | .3637 | 18.80 | .6428 | .3703 | 17.68 | .6238 | .3319 | 18.76 | .6512 | .3991 |
| Ours | 21.28 | .5337 | .3562 | 19.72 | .4848 | .3595 | 21.31 | .5264 | .4698 | 19.64 | .6107 | .2942 | 17.11 | .5388 | .3313 | 19.68 | .6007 | .3774 | 19.17 | .6600 | .3577 | 17.93 | .6324 | .3248 | 18.93 | .6602 | .3970 |

Table 5: **Per-dataset results of novel view synthesis metrics on Feat2GS benchmark.** Results indicate that our encoder leads to the best performance on most datasets in Geometry, Texture, and All probing modes. The highest, second-highest, and third-highest scores in each category are highlighted with  light red ,  light orange , and  light yellow , respectively.

| Method | NYU-v2 (Single-frame) | | BONN (Video) | |
|---|---|---|---|---|
| | Abs Rel ↓ | $\delta < 1.25$ ↑ | Abs Rel ↓ | $\delta < 1.25$ ↑ |
| DUSt3R | 0.080 | 90.7 | 0.155 | 83.3 |
| MonST3R | 0.102 | 88.0 | **0.067** | **96.3** |
| Spann3R | 0.122 | 84.9 | 0.144 | 81.3 |
| MASt3R | 0.129 | 84.9 | 0.252 | 70.1 |
| Ours | **0.070** | **93.9** | 0.086 | 91.4 |

| Method | Params | ImageNet1K | | Segmentation | |
|---|---|---|---|---|---|
| | (M) | Zero-shot | k-NN | ADE20k | VOC |
| SAM-H/16 | 637 | - | 22.12 | 28.08 | 34.34 |
| OpenAI CLIP-L/14 | 305 | 75.54 | 79.96 | 36.51 | 67.04 |
| SigLIP-L/14 | 428 | **82.61** | **85.16** | 40.53 | 70.31 |
| DINOv2-g/14-reg | 1,137 | - | 83.41 | 48.68 | 82.78 |
| DUSt3R Enc. | 303 | - | - | 32.10 | 46.02 |
| DUNE-B/14-448 | 420 | - | - | 45.60 | - |
| MASt3R Enc. | 303 | - | - | 32.54 | 48.58 |
| ∗RADIOv2.5-L | 320 | 80.55 | 83.16 | **50.68** | **85.60** |
| *UniUGG* Enc. (Ours) | 320 | 80.06 | 83.13 | 50.12 | 85.43 |

Table 6: *Left:* **Depth evaluation results.** We report single-frame depth evaluation performance on the NYU-v2 dataset and video depth evaluation performance on the BONN dataset. *Right:* **Comparison of encoder performance on the image/pixel level.** 'Zero-Shot' and k-NN are computed on ImageNet-1K. ADE20K and PascalVOC2012 refer to linear probe semantic segmentation mIOU. ∗denotes teachers used to pretrain our encoder.

encoder (Wang et al., 2024b),DUNE-B/14-448 (Sarıyıldız et al., 2025) ,MASt3R encoder (Leroy et al., 2024) and RADIOv2.5-L (Ranzinger et al., 2024b).

Following (Ranzinger et al., 2024b), we freeze the vision encoders and train a linear head on top of the frozen features. The linear probe is conducted in the MMSeg (Contributors, 2020) framework. We train the linear head for 20k steps using a total batch size of 64, a base learning rate of $5e^{-3}$, and the Adam-W optimizer.

As shown in Tab. 6 (right), while our method does not outperform the teacher baseline, it yields competitive results that validate the potential of our encoder. More importantly, it establishes a strong foundation for unified spatial reasoning and 3D generation.

**Downstream task performance** As shown in Tab. 7, we provide more detailed results about *Tab. 2 of the main text*. Our encoder (*UniUGG* Enc.) demonstrates clear advantages on spatial reasoning tasks and two general QA benchmarks. It shows that our encoder has more consistent performance across modalities, balancing spatial perception and high-level semantics.

### A.3.2 EVALUATION OF SPATIAL UNDERSTANDING

**Detailed spatial understanding scores** As shown in Tab. 8, we present more fine-grained spatial understanding scores on 3DSRBench and VSI-Bench (corresponding to *Tab. 3 of the main text*). We jointly train *UniUGG* for both spatial understanding and 3D generation tasks. The model utilizes our pretrained geometric-semantic encoder as the visual backbone and employs Qwen2.5-3B-Instruct as the large language model. The results demonstrate that *UniUGG* can capture precise spatial relations by jointly modeling 3D structure and visual-language reasoning. It should be noted that the LLM used in *UniUGG* has a size of only 3B parameters.

| Method | Params | VSI-Bench | | | | | | | | SPAR | | |
|---|---|---|---|---|---|---|---|---|---|---|---|---|
| | (M) | Count | Obj.Size | Room | Rel.Dir. | Rel.Dist. | Route | Order | Avg. | Low | Medium | High |
| Qwen25-ViT | 669 | 58.00 | 51.12 | 31.04 | 37.31 | 29.44 | 25.26 | 21.20 | 35.56 | 36.50 | 36.67 | 39.89 |
| OpenAI CLIP-L/14 | 305 | 61.43 | 49.30 | 49.31 | 39.45 | 36.20 | 30.93 | 16.18 | 40.08 | 44.13 | 43.67 | 52.33 |
| SigLIP-L/16 | 316 | 11.93 | 42.06 | 0.73 | 38.95 | 27.89 | 28.87 | 9.71 | 23.81 | 41.75 | 34.67 | 43.11 |
| MASt3R Encoder | 303 | 58.12 | 39.81 | 48.82 | 43.70 | 33.24 | 32.47 | 19.58 | 39.14 | 50.00 | 42.33 | 48.22 |
| RADIOv2.5-L | 320 | 59.91 | 53.49 | 50.17 | 37.08 | 32.54 | 30.41 | 16.18 | 39.75 | 50.44 | 47.95 | 52.13 |
| *UniUGG* Enc. (Ours) | 320 | 62.69 | 51.70 | 49.34 | 42.14 | 34.37 | 32.99 | 27.51 | 42.18 | 50.82 | 49.07 | 51.89 |

| Method | Params | BLINK | | | | | | | | | Seed-I | Real World |
|---|---|---|---|---|---|---|---|---|---|---|---|---|
| | (M) | Fun.Corr. | Vis.Corr. | Local. | Jigsaw | Depth | Spatial | Simi. | Art | Avg. | | |
| Qwen25-ViT | 669 | 15.38 | 26.16 | 54.92 | 46.67 | 44.35 | 52.45 | 46.67 | 52.99 | 37.87 | 41.81 | 44.97 |
| OpenAI CLIP-L/14 | 305 | 24.62 | 24.42 | 52.46 | 56.00 | 45.97 | 63.64 | 43.70 | 52.99 | 40.45 | 69.14 | 54.38 |
| SigLIP-L/16 | 316 | 20.00 | 20.35 | 61.48 | 51.33 | 46.77 | 53.85 | 53.33 | 56.41 | 39.08 | 56.31 | 45.23 |
| MASt3R Encoder | 303 | 22.31 | 27.33 | 55.74 | 48.67 | 44.35 | 67.13 | 44.44 | 45.30 | 40.93 | 56.96 | 50.07 |
| RADIOv2.5-L | 320 | 23.08 | 31.98 | 47.54 | 43.33 | 68.55 | 65.53 | 47.31 | 53.85 | 42.92 | 72.09 | 57.38 |
| *UniUGG* Enc. (Ours) | 320 | 33.85 | 33.14 | 55.74 | 48.67 | 69.35 | 67.83 | 51.11 | 59.85 | 44.40 | 71.65 | 58.56 |

Table 7: **Detailed comparison results of encoder performance on downstream vision-language reasoning benchmarks.** The VLM architecture is based on Qwen2.5-3B-Instruct, and all encoders are trained under the same settings to ensure fairness.

| Method | 3DSRBench | | | | | VSI-Bench | | | | | | | | |
|---|---|---|---|---|---|---|---|---|---|---|---|---|---|---|
| | Height | Loc. | Orient. | Multi. | Avg. | Obj.Count | Abs.Dist | Obj.Size | RoomSize | Rel.Dist | Rel.Dir | RoutePlan | Appr.Order | Avg. |
| LLaVA-v1.5-7B | 39.1 | 46.9 | 28.7 | 34.7 | 38.1 | 6.2 | 4.9 | 32.6 | 2.7 | 29.6 | 30.7 | 26.3 | 10.5 | 18.0 |
| LLaVA-NeXT-7B | 50.6 | 59.9 | 36.1 | 43.4 | 48.4 | 7.5 | 8.8 | 27.7 | 25.8 | 33.2 | 29.7 | 23.7 | 8.6 | 20.6 |
| InternVL2.5-8B | 45.9 | 68.1 | 38.7 | 43.3 | 50.9 | 7.7 | 32.6 | 42.9 | 34.6 | 39.6 | 40.0 | 24.7 | 37.7 | 32.5 |
| Qwen2.5-VL-7B | 44.1 | 62.7 | 40.6 | 40.5 | 48.4 | 26.7 | 10.8 | 35.4 | 31.0 | 35.2 | 38.2 | 35.1 | 29.6 | 30.3 |
| GPT-4o | 53.2 | 59.6 | 21.6 | 39.0 | 44.2 | 46.2 | 5.3 | 43.8 | 38.2 | 37.0 | 41.3 | 31.5 | 28.5 | 34.0 |
| *UniUGG*-3B (Ours) | 52.3 | 60.0 | 43.4 | 49.3 | 52.1 | 63.2 | 34.8 | 49.1 | 50.4 | 30.3 | 38.6 | 26.3 | 26.7 | 40.1 |

Table 8: **Detailed spatial understanding scores on 3DSRBench and VSI-Bench.** Our *UniUGG* is jointly trained for both spatial understanding and 3D generation tasks. Note that the LLM used in *UniUGG* has a size of only 3B parameters.

**Evaluation on 3D-specific benchmarks** We further evaluate the 3D reasoning capability of our *UniUGG* on SQA3D (Ma et al., 2022), ScanQA (Azuma et al., 2022), and ScanRefer (Chen et al., 2020), which are widely adopted benchmarks in 3D VLMs. Due to the differences in QA formats, and following the practice of prior works, we apply supervised fine-tuning on 3D tasks to better assess *UniUGG*'s spatial capabilities. First, we evaluate spatial understanding using SQA3D and ScanQA, which focus on object attributes, spatial relations, and viewpoint-conditioned reasoning. Compared models include 3D-LLM (Hong et al., 2023), Chat-3D v2 (Huang et al., 2023a), LEO (Huang et al., 2023b), LL3DA (Chen et al., 2024e), Scene-LLM (Fu et al., 2024a), LLaVA-3D (Zhu et al., 2024a) and Video-3D LLM (Zheng et al., 2024). As shown in Tab. 9 (left), although our model does not incorporate explicit 3D information, it achieves competitive performance compared to 3D-enhanced LLMs. Additionally, we assess 3D grounding on the ScanRefer dataset, where the task requires localizing target objects based on textual descriptions. Compared models include ScanRefer, MVT (Huang et al., 2022), ViL3DRel (Chen et al., 2022), 3D-LLM, Chat-3D v2, Grounded 3D-LLM (Chen et al., 2024f), LLaVA-3D, and Video-3D LLM. As shown in Tab. 9 (right), our model may struggle to predict depth accurately due to the absence of 3D information, resulting in poor grounding results. To address this, we apply a refinement strategy to mitigate this issue and improve grounding performance. These results highlight the potential of our *UniUGG* to perform spatial reasoning and grounding without relying on explicit 3D inputs, while still supporting more complex generative tasks. It should be noted that our model employs a backbone with 3B parameters, while other methods, for example, LLaVA-3D and Video-3D-LLM, use 7B-parameter backbones.

A.3.3 EVALUATION OF 3D GENERATION

**More qualitative results of 3D generation** We provide more visualization results to further demonstrate the 3D generation and understanding capabilities of *UniUGG*. Given a reference image, we randomly sample plausible relative view transformations and let *UniUGG* generate the corresponding 3D scenes. *UniUGG* further captioned the generated 3D scenes. As shown in Fig. 7, Fig. 8, Fig. 9,

| Method | 3D | SQA3D | ScanQA | | |
| --- | --- | --- | --- | --- | --- |
| | | EM | BLEU-4 | CiDEr | EM |
| 3D-LLM | ✓ | - | 12.0 | 69.4 | 20.4 |
| Chat-3D v2 | ✓ | 54.7 | 14.0 | 87.6 | - |
| LEO | ✓ | 50.0 | 13.2 | 101.4 | 21.5 |
| LL3DA | ✓ | - | 13.5 | 76.8 | - |
| Scene-LLM | ✓ | 54.2 | 12.0 | 80.0 | 27.2 |
| LLaVA-3D | ✓ | 55.6 | 14.5 | 91.7 | 27.0 |
| Video-3D LLM | ✓ | **58.6** | **16.2** | **102.1** | **30.1** |
| *UniUGG* (Ours) | ✗ | 51.3 | 12.1 | 85.6 | 24.4 |

| Methods | Acc@0.25 | Acc@0.5 |
| --- | --- | --- |
| ScanRefer | 37.3 | 24.3 |
| MVT | 40.8 | 33.3 |
| ViL3DRel | 47.9 | 37.7 |
| 3D-LLM | 30.3 | - |
| Chat-3D v2 | 35.9 | 30.4 |
| Grounded 3D-LLM | 47.9 | 44.1 |
| LLaVA-3D | 54.1 | 42.4 |
| Video-3D LLM | **58.1** | **51.7** |
| ∗*UniUGG* (Ours) | 23.2 | 7.8 |
| *UniUGG* (Ours) | 41.9 | 36.6 |

Table 9: *Left:*Performance comparison on SQA3D and ScanQA. "3D" indicates whether the model is infused with 3D information. *Right:*Performance comparison across different models on ScanRefer. ∗indicates results without refine.

| Method | VSI | BLINK | 3DSR | SPAR |
| --- | --- | --- | --- | --- |
| *UniUGG* (full) | 40.1 | 43.6 | **52.1** | 47.2 |
| *UniUGG* (und. only) | **42.2** | **44.4** | 51.3 | **49.8** |

| Method | ARKitScenes | | | ScanNet++ | | |
| --- | --- | --- | --- | --- | --- | --- |
| | FID↓ | KID↓ | LPIPS↓ | FID↓ | KID↓ | LPIPS↓ |
| *UniUGG* (full) | 55.01 | .0425 | .4849 | 55.64 | **.0442** | .4263 |
| *UniUGG* (gen. only) | **54.35** | **.0409** | .4556 | 55.46 | .0454 | **.3586** |

Table 10: **Performance comparison with separate optimization.**

and Fig. 10, *UniUGG* consistently produces geometrically coherent and semantically meaningful 3D content, along with accurate scene captions.

**Feature matching visualization** By taking only a reference image and the relative view transformation as input, *UniUGG* predicts the ViT tokens of the target image, which, together with the reference tokens, are then decoded into point clouds and the corresponding cross-view feature matchings. We visualize these feature matching results in Fig. 11 and compare them with the baseline method MASt3R (Leroy et al., 2024), which requires both the reference and ground-truth transformed-view images as input to obtain feature matching. Our method achieves highly accurate feature matching, which is highly consistent with those produced by MASt3R. This demonstrates that *UniUGG* not only generates the spatial structure under the novel view but also ensures that the generated content is generally consistent with the input view transformation condition.

**Robust generation under extreme view transformation** We illustrate the generation performance of *UniUGG* under extreme view transformation conditions, as shown in Fig. 12. We evaluate the model performance conditioned on rotation angles of 60°, 80°, 100°, 120°, and 140°. As observed, within the range of 60°-120°, *UniUGG* still produces high-quality, semantically rich, and structurally coherent 3D point clouds. However, at 140° or beyond, the generation quality drops noticeably, with the point clouds exhibiting blurred textures and distorted structures. This degradation arises from two factors. First, during training, the view-transformation conditions are usually constrained within: view overlap ratio > 0.4, rel-translation < 2 m, and rel-rotation < 120°. Consequently, when evaluated under extreme view transformation (e.g., rotation > 140°), our model struggles to maintain generation quality. Second, such large viewpoint changes significantly reduce the overlap between the reference and target images, resulting in insufficient cross-view constraints and finally leading to failures in 3D generation.

**Failure case** Due to the lack of training samples with large view transformations, *UniUGG* would generate point clouds with blurred textures and distorted structure under extreme viewpoint changes, as shown in Fig. 12. What's more, we also provide some examples of failure cases in Fig. 13. In some cases, *UniUGG* generates point clouds with color distortions, where vivid green regions are mistakenly inferred as grayish colors. This may be caused by a conflict between color decoding and semantic representation tasks during encoder pretraining.

### A.3.4 PERFORMANCE OF SEPARATELY OPTIMIZED

As shown in Tab. 10, we separately optimize the spatial understanding task and the 3D generation task, and compare their performance with that of the jointly trained model. Benefiting from the Spatial-VAE and diffusion model, joint training (*UniUGG* full) does not significantly degrade the performance of 3D generation (*UniUGG* gen. only) and only leads to a moderate reduction in

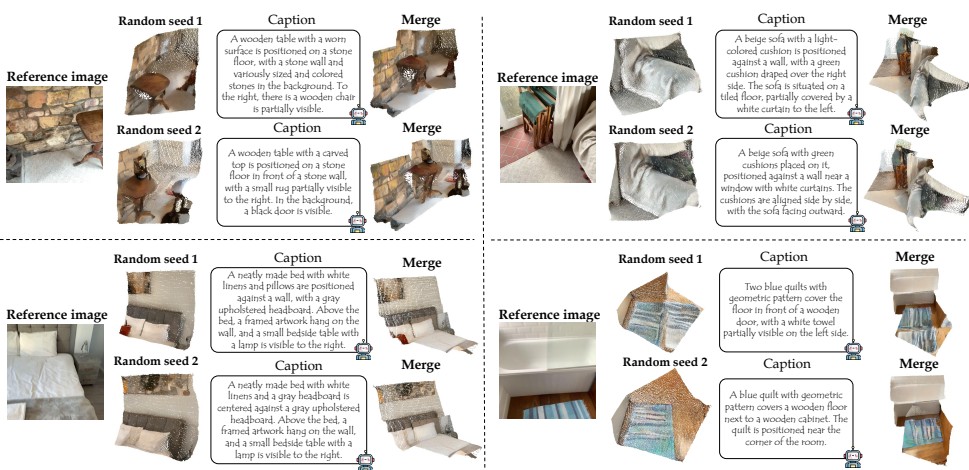

Figure 7: **Additional visualization samples.** We present 3D scene generations and the corresponding captions produced by our *UniUGG* under varying random seeds.

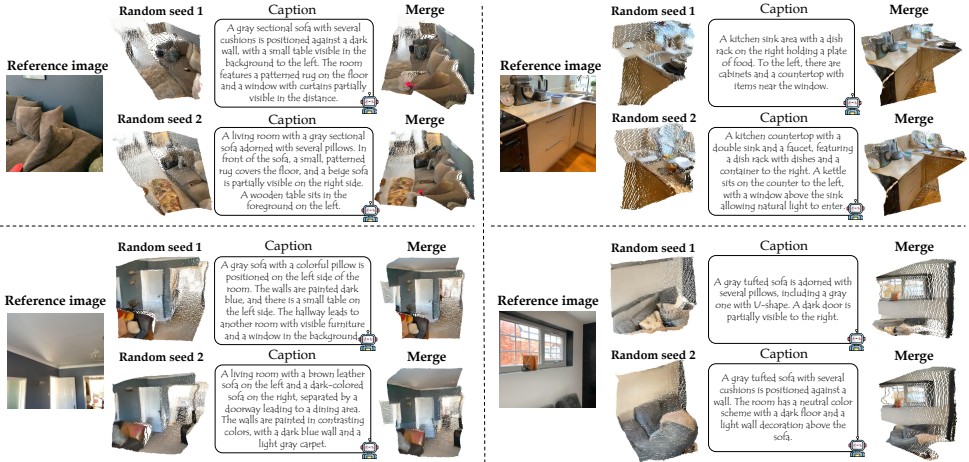

Figure 8: **Additional visualization samples.** We present 3D scene generations and the corresponding captions produced by our *UniUGG* under varying random seeds.

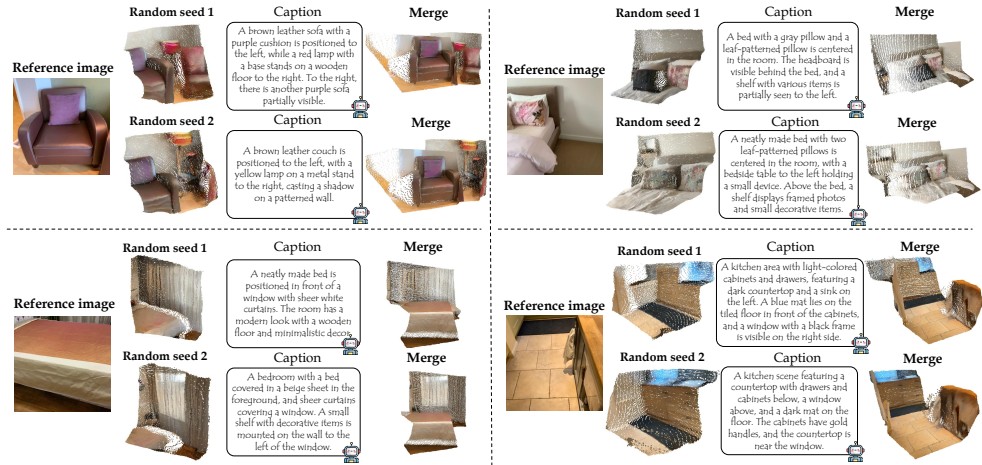

Figure 9: **Additional visualization samples.** We present 3D scene generations and the corresponding captions produced by our *UniUGG* under varying random seeds.

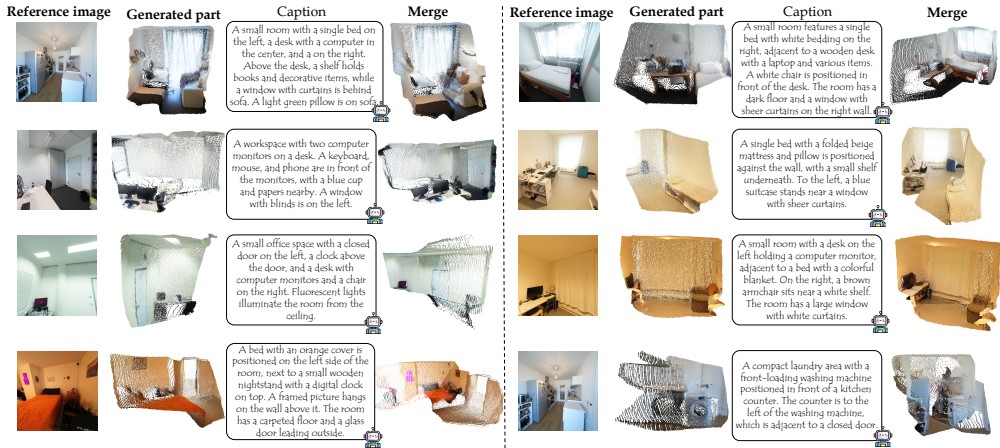

Figure 10: **Additional visualization samples.** We present 3D scene generations and the corresponding captions produced by our *UniUGG*.

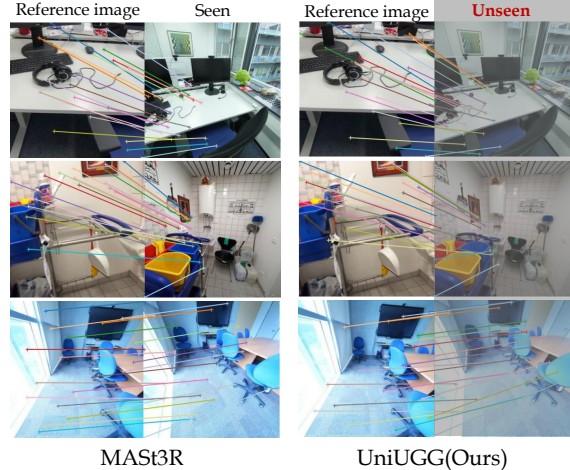

Figure 11: **Feature matching visualization.** Without access to the target image, our method takes the reference image and the relative view transformation as input and still produces accurate feature matchings, while baseline MASt3R takes image pairs as input.

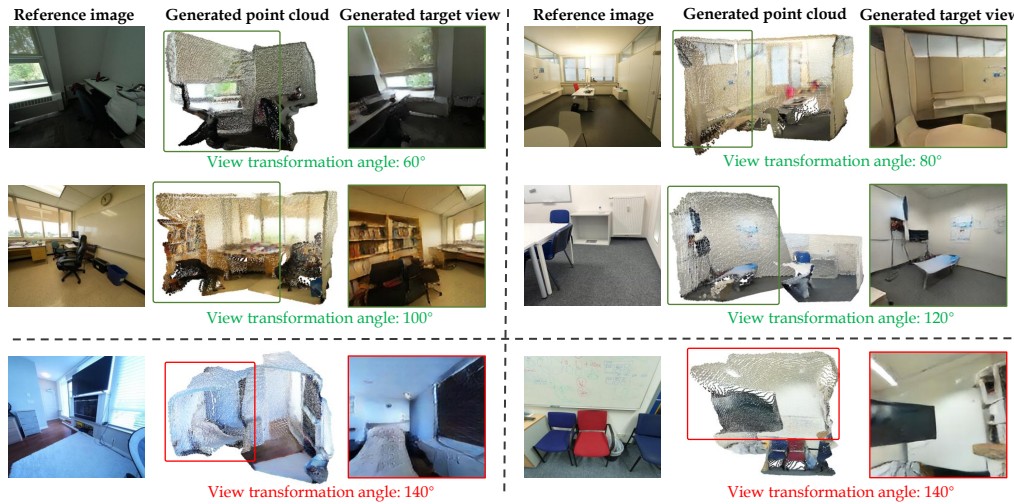

Figure 12: **3D generation under extreme view transformation.** When the rotation angle is below 120°, *UniUGG* can still produce high-quality 3D point clouds under the target viewpoint. As the rotation angle increases to 140° or beyond, the quality of the generated results degrades noticeably.

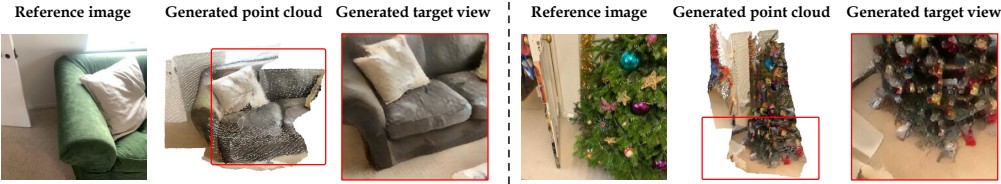

Figure 13: **Other failure cases of our model.**

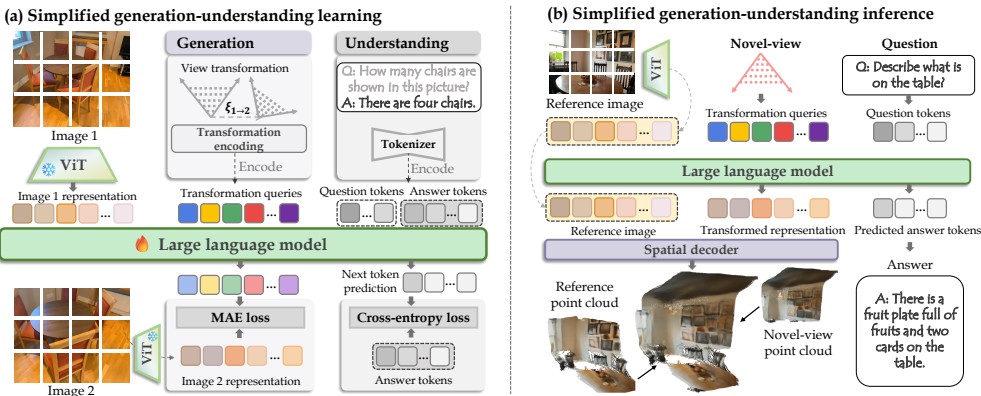

Figure 14: **Overview of our simplified model variant pipeline.** In the simplified variant, we do not use the diffusion model and Spatial-VAE for scene generation but directly produce the novel-view tokens by LLM, which is supervised with ground truth tokens.

spatial understanding (*UniUGG* und. only) performance. We believe this performance trade-off is acceptable: although a small amount of understanding performance is sacrificed, we obtain a unified 3D model for both understanding and generation. This unification substantially reduces training cost and computational overhead while broadening the range of tasks the model can support.

## A.4 EXPLORATION OF SIMPLIFIED MODEL VARIANT

During the development of our proposed *UniUGG*, we explored the early or simplified version of the model architecture. This helped us better understand the role of the core component in the model. Although not intended as the final model, the simplified variant provides valuable insights for designing a unified framework for 3D understanding and generation.

As shown in Fig 14 (a), during training, LLM learns both spatial generation capability across views by predicting ViT tokens of the novel view, and visual understanding capability via question answering As shown in Fig 14 (b), at inference, the model performs either novel-view spatial generation or question answering using a single image as reference, conditioned on the input type.

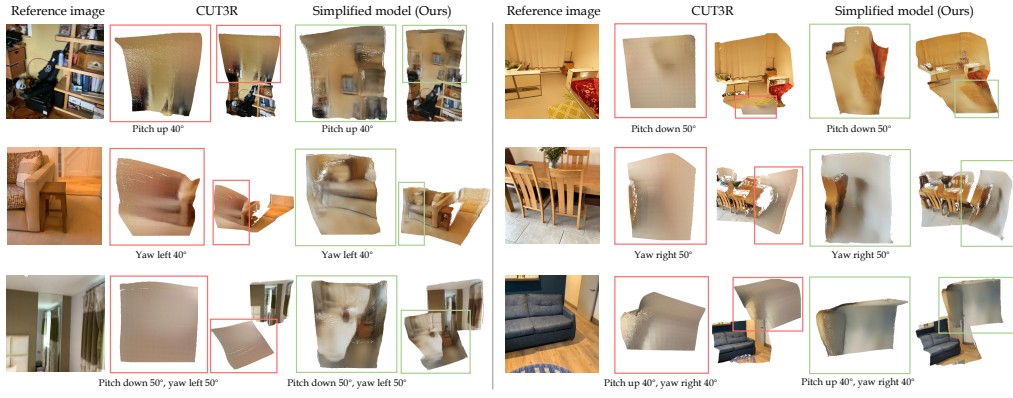

Figure 15: **Visualization of scenes generated by the simplified model variant.** Although the simplified variant can imagine spatial structures under novel views, it lacks clear details.

We present visualization generation results of the simplified variant in Fig. 15. While it outperforms the baseline CUT3R in terms of generating fine-grained spatial structures, its textures remain noticeably inferior to those of our full *UniUGG*. This performance gap is primarily attributed to the absence of the diffusion prior and Spatial-VAE, which play a crucial role in modeling high-frequency visual details and realistic textures. The simplified variant, lacking these core components, tends to generate over-smoothed surfaces.

