# OpenReview forum: "UniUGG: Unified 3D Understanding and Generation via Geometric-Semantic Encoding"
_ICLR.cc/2026/Conference — ICLR 2026 Poster_

### Official Review · Reviewer_4To6 · 2025-10-29

**Soundness:** 3
**Presentation:** 1
**Contribution:** 2
**Rating:** 4
**Confidence:** 3

**Summary:**

This paper presents UniUGG, which is a unified model for both 3D understanding and generation. Specifically, both geometric and semantic information of input images is encoded with the visual encoder. Also, a spatial decoder is used for 3D generation. Experimental results show that the proposed unified 3D understanding and generation method achieves superior performance for the visual representation of the visual encoder, spatial understanding capability, and 3D generation capability.

**Strengths:**

- Having a unified model for 3D understanding and generation is pioneering, which can probably serve as the basic work for the unified 3D model direction.

- The idea of binding a MASt3R-powered spatial decoder with LLMs is innovative and reasonable to me, which enables the whole system to output something that is spatial-aware, like the point cloud in the generation part of this work.

- The proposed method achieves decent experimental results on the benchmarks for visual representation, spatial understanding, and 3D generation, demonstrating that it is a well-rounded unified 3D model.

**Weaknesses:**

- It remains unanswered for the motivation of proposing such a unified 3D foundation model for both understanding and generation. In other words, there is lack of evidence that modeling them together in a single model can provide mutual benefit to each other. If there is no mutual benefit between these two streams of tasks, or even slightly harm each other, it will become unjustified for us to design such a delicate framework for unifying understanding and generation for 3D, because optimizing them separately may produce optimal performance.

- I think it is a bit overclaiming to say the understanding task UniUGG solves are 3D understanding. Most of the tasks (like the ones in VSI-Bench, BLINK) are the vision-language tasks related to spatial reasoning, with limited relation with 3D understanding. I would say the benchmarks like SQA3D [1], ScanQA [2], ScanRefer [3] are more related to 3D understanding, as they are understanding and reasoning with 3D representations, instead of only reasoning on single (or sometimes multi-view) images for some spatial understanding tasks.

- The writing of the methodology section (Sec. 3) makes me feel a bit difficult to understand how the proposed UniUGG framework looks like. It starts with introducing the visual encoder, which I find a bit lost because I even do not have a rough idea in mind about what modules the full UniUGG framework has. Therefore, I find confusing where is this visual encoder locates in the whole framework. The introduction of the following modules also share the same problem. Thus, I believe this section lacks a part that introduces the rough whole pipeline to audience at the beginning, for them to have a big picture in mind.

- The pipeline figures (Figures 2-4) are also not clear enough. They are crowded with small text, and are hard to understand. Basically, I cannot find a clear figure showing what the inputs and outputs are for the model. The contents inside the figure are also confusing. For example, in Figure 3(b), the large language model seems to take in image 2 representation together with transformation queries and question & answer tokens. Is image 1 the target image that we want to generate, so it is not processed through large language models. And the encoding of image 1 and image 2 seems to be messed up with the representation of image 1 and image 2.

[1] Ma et al. SQA3D: Situated Question Answering in 3D Scenes. ICLR 2023.

[2] Azuma et al. ScanQA: 3D Question Answering for Spatial Scene Understanding. CVPR 2022.

[3] Chen et al. ScanRefer: 3D Object Localization in RGB-D Scans using Natural Language. ECCV 2020.

**Questions:**

- How will the model perform if understanding and generation tasks are separately optimized? If separately optimizing them can achieve better performance for each of them, then there is limited justification for modeling them together. Otherwise, the cases (e.g., interactive multi-round scene editing like those in Uni3D-LLM [1]) that requires joint understanding and generation capabilities need to be shown.

- How will the model perform in the more perceived 3D understanding / reasoning tasks, like SQA3D, ScanQA, ScanRefer that are more used in 3D VLMs?

- What is the training cost of this work? The paper only mentions about the batch size and the number of optimization steps of the training the model, without mentioning what kind of GPU resources they are using, and how much time is needed for training the model, which makes it unclear how efficient this algorithm is.

[1] Liu et al. Uni3D-LLM: Unifying Point Cloud Perception, Generation and Editing with Large Language Models. arXiv:2402.03327.

---

> ### Author Response · Authors · 2025-11-22
> **Response to reviewer 4To6 (1/3)**
>
> Thanks for the positive and detailed review as well as the valuable suggestions for improvement. We would like to address the reviewer's concerns as follows:
>
> **W1: Motivation for unified models.**
>
>
> Thanks for the insightful question. As discussed in our introduction, an increasing number of recent studies (e.g., Janus[1], Show-o[2], ILLUME[3]) have attempted to explore unified models for visual understanding and generation. However, these efforts are limited to the 2D domain, while unified 3D models remain largely unexplored. Our work **aims to fill this gap by proposing a unified generation and understanding framework for 3D scenes**. In response to the reviewer's concern, we articulate our motivation from the following perspectives:
>
> (1) **Shared visual representations.**
> Spatial understanding and 3D generation fundamentally rely on a common geometric-semantic representation. Understanding focuses more on global, high-level semantics, while generation emphasizes detailed local geometry and textures. A unified training paradigm allows the model to learn these complementary representations simultaneously, resulting in stronger cross-task generalization than training the tasks separately.
>
> (2) **Balanced performance.**
> In our response to Q1, we present the performance of models trained solely on the understanding or generation task. The results indicate that joint training does not significantly degrade 3D generation performance and only causes a moderate drop in spatial understanding accuracy. We believe this performance trade-off is acceptable, as the unified model enables capabilities that two separate models cannot achieve. For example, controllable 3D scene generation based on complex instructions requires reliable spatial understanding and grounding capability to correctly interpret user intent and object spatial relationships.
>
> (3) **Multimodal applications.**
> Real-world 3D applications are rarely confined to “understanding only” or “generation only.” Many practical scenarios, eg, multi-round 3D scene editing, AR/VR interactive creation, require the model to operate within an understanding-reasoning-generation closed-loop. This makes the unified architecture a naturally appropriate solution. Although our current system still has limitations (e.g., it does not yet support multi-round 3D interactive editing, as discussed in Section 5), building a unified 3D model extends LLMs to a wide range of multimodal applications and unlocks new possibilities for improving the synergy between vision and language tasks.
>
> (4) **Potential mutual benefits**.
> Although understanding and generation have different objectives, many unified 2D models[3][4] have demonstrated that these tasks can benefit each other through shared latent spaces and complementary supervision.
>
> (5) **Reduced parameters and training cost.**
> Building a unified model rather than two separate systems substantially reduces model parameters and deployment cost, as both tasks share the same ViT encoder and LLM backbone. Moreover, joint training is computationally more efficient: as noted in response to Q3, our unified training in stage 3 requires 58 hours on a cluster with 8 nodes, each equipped with 8 Ascend NPUs, whereas training the understanding-only model in stage 3 takes 50 hours on the same setup, and the generation-only model in stage 3 requires 48 hours on 8×A6000 GPUs. (Stages 1 and 2 remain unchanged regardless of whether the tasks are trained jointly or separately.)
>
> In summary, although unified 3D modeling is still in its early stage, we believe that a unified model for both spatial understanding and 3D generation is not only feasible but also beneficial and practically valuable, **with clear advantages in capability, generalization, and efficiency.**

---

> ### Author Response · Authors · 2025-11-22
> **Response to reviewer 4To6 (2/3)**
>
> **W2: Evaluation on other 3D benchmarks.**
>
> Thanks for the valuable suggestion. Following the reviewer's advice, we further evaluated our model on SQA3D, ScanQA, and ScanRefer, which are widely used benchmarks for spatial reasoning in 3D VLMs. The complete results and corresponding analysis have been added to **Appendix A.3.2 and Table 9** of the revised manuscript. The results are summarized as follows:
>
>
> | Method         | 3D | SQA3D| ScanQA | | |
> |----------------|:----------:|:--------------:|:----------:|:--------------:|:--------------:|
> |  |   |  EM | BLEU-4     | CiDEr    | EM      |
> | 3D-LLM         | ✓  | -        | 12.0           | 69.4          | 20.4      |
> | Chat-3D v2     | ✓  | 54.7     | 14.0           | 87.6          | -         |
> | LEO            | ✓  | 50.0     | 13.2           | 101.4         | 21.5      |
> | LL3DA          | ✓  | -        | 13.5           | 76.8          | -         |
> | Scene-LLM      | ✓  | 54.2     | 12.0           | 80.0          | 27.2      |
> | LLaVA-3D       | ✓  | 55.6     | 14.5           | 91.7          | 27.0      |
> | Video-3D LLM   | ✓  | **58.6** | **16.2**       | **102.1**     | **30.1**  |
> | **UniUGG (Ours)** | ✗  | 51.3     | 12.1           | 85.6          | 24.4      |
>
> | Methods           | ScanRefer |  |
> |-------------------|:--------------:|:--------------:|
> |   | Acc@0.25 | Acc@0.5 |
> | ScanRefer         | 37.3      | 24.3     |
> | MVT               | 40.8      | 33.3     |
> | ViL3DRel          | 47.9      | 37.7     |
> | 3D-LLM            | 30.3      | -        |
> | Chat-3D v2        | 35.9      | 30.4     |
> | Grounded 3D-LLM   | 47.9      | 44.1     |
> | LLaVA-3D          | 54.1      | 42.4     |
> | Video-3D LLM      | **58.1**  | **51.7** |
> | **UniUGG (Ours)** | 41.9      | 36.6   |
>
>
> Due to the differences in QA formats, and following common practice in prior work, we perform supervised fine-tuning on the 3D tasks to better evaluate UniUGG’s spatial capabilities. As shown in the Table above, despite **not utilizing explicit 3D information**, our model achieves performance competitive compared to 3D-enhanced LLMs.
> It should be noted that our model employs a **backbone with 3B parameters**, while other methods, for example, LLaVA-3D[5] and Video-3D-LLM[6], use 7B-parameter backbones.
> These results demonstrate the strong potential of UniUGG to conduct spatial reasoning and grounding without relying on explicit 3D inputs, while still supporting more complex 3D generative tasks.
>
> **W3: Lack of overall pipeline overview.**
>
> Thanks for the valuable suggestion. We have **added Section 3.1** in the revised manuscript to provide a clearer and more structured introduction to the overall workflow of our proposed UniUGG. We have also **included an overarching and highly simplified overview diagram** (Figure 2 in the revision) to help readers better grasp the key components and the overall pipeline.
>
> As shown in Figure 2 of the revised manuscript, our training pipeline follows a **three-stage strategy**: (1) Pretrain the vision encoder to learn geometric-semantic representations, (2) Pretrain the Spatial-VAE to compress representations, and (3) Jointly train the UniUGG for spatial reasoning and 3D generation, with the vision encoder and VAE encoder frozen. Therefore, **the visual encoder mentioned** in the comment is first pretrained and then used to extract geometric–semantic representations from images, which will be fed into LLM.
>
> **W4: Unclear and confusing pipeline figures.**
>
> Thanks for pointing out this issue. In Figure 3(b) (renamed as Figure 4(b) in the revised manuscript), the encoding of Image 1 and Image 2 was messed up with their representations. We have **corrected this error** in the revised version.
>
> Following the reviewer's suggestion, we have also **refined Figures 2-4** (renamed as Figures 3-5 in the revision) by improving the layout and enlarging the font sizes to enhance readability. In addition, we added Section 3.1 and an overall pipeline diagram to clearly illustrate the inputs, outputs, and workflow of UniUGG. We hope these updates can address the reviewer's concerns.

---

> ### Author Response · Authors · 2025-11-22
> **Response to reviewer 4To6 (3/3)**
>
> **Q1: Performance of separately optimized.**
>
> Thanks for the question. We have added experiments that separately train the understanding task and the 3D generation task. **The results are reported as follows**:
> | Method    | ARKitScenes       |        |         | ScanNet++         |        |         |
> |--------------|:--------------------:|:---------------------:|:------------------:|:-----------------------------:|:--------------------:|:-----------------------:|
> |    | FID↓ | KID↓   | LPIPS↓ |  FID↓ | KID↓   | LPIPS↓ |
> |CUT3R | 138.54 |0.1128 |0.5758 | 130.76 |0.1051 |0.5637  |
> |LVSM | 269.45 |0.3088 |0.5067 | 414.63 |0.5117 |0.5865 |
> | UniUGG (full) | 55.01   | 0.0425  | 0.4849  | 55.64  | **0.0442**  | 0.4263  |
> | **UniUGG (gen. only)** | **54.35**     | **0.0409**  | **0.4556**  |**55.46**       | 0.0454  | **0.3586**  |
>
> | Method    | VSI       |   BLINK     |     3DSR    | SPAR|
> |--------------|:-----------------:|:---------------:|:--------:|:--------------:|
> | UniUGG (full) | 40.1| 43.6| **52.1**|47.2|
> | **UniUGG (und. only)**| **42.2**     | **44.4**  | 51.3  |**49.8** |
>
> Benefiting from the Spatial-VAE and diffusion model, **joint training does not significantly degrade the performance of 3D generation and only leads to a moderate reduction in spatial understanding performance**. We believe **this performance trade-off is acceptable**: although a small amount of understanding performance is sacrificed, we obtain a unified 3D model for both understanding and generation. This unification substantially reduces training cost and computational overhead while broadening the range of tasks the model can support.
>
> As discussed in Section 5, our model still has certain limitations and is currently unable to support multi-round scene generation and editing. However, **unlike Uni3D-LLM[7], which primarily focuses on object-level generation and then inserts the object into the scene, our approach emphasizes full scene-level point cloud generation.** We believe that complex interactive scene-generation tasks require a model with strong spatial perception, enabling it to correctly interpret and execute spatially grounded generation instructions.
>
>
> **Q2: Performance on 3D-specific benchmarks.**
>
> Thanks for the comment. In response, we evaluated our model on SQA3D, ScanQA, and ScanRefer. The complete results and corresponding analysis have been added to **Appendix A.3.2 and Table 9** of the revised manuscript (See response to W2).
>
> **Q3: Training efficiency and resource cost.**
>
> Thanks for the suggestion. We have reported the training cost and computational resources in **Section 4.1** of the revised manuscript. Our pipeline adopts a three-stage training strategy as illustrated in Figure 2. In stage 1, we pretrain the geometry-semantic encoder for 25 hours using 8× NVIDIA A6000 GPUs. In stage 2, the Spatial-VAE module is pretrained for 12 hours on the same 8× A6000 GPUs. In stage 3, we train the full UniUGG model on both spatial understanding and 3D generation tasks for about 58 hours on a cluster with 8 nodes, each equipped with 8 Ascend NPUs. During inference, we run our model on a single NVIDIA A6000 GPU. For spatial reasoning, given 16 input images at a resolution of 224×224, UniUGG achieves an inference latency of 350ms, utilizing bf16 precision and FlashAttention for acceleration. For 3D generation, UniUGG takes approximately 1.2s to generate a point cloud from a single reference image.
>
> >[1] Chengyue Wu, Xiaokang Chen, Zhiyu Wu, Yiyang Ma, Xingchao Liu, Zizheng Pan, Wen Liu, Zhenda Xie, Xingkai Yu, Chong Ruan, et al. Janus: Decoupling visual encoding for unified multimodal understanding and generation. In CVPR, 2025.
>
> >[2] Jinheng Xie, Weijia Mao, Zechen Bai, David Junhao Zhang, Weihao Wang, Kevin Qinghong Lin, Yuchao Gu, Zhijie Chen, Zhenheng Yang, and Mike Zheng Shou. Show-o: One single transformer to unify multimodal understanding and generation. In ICLR, 2025
>
> >[3] Chunwei Wang, Guansong Lu, Junwei Yang, Runhui Huang, Jianhua Han, Lu Hou, Wei Zhang, and Hang Xu. Illume: Illuminating your llms to see, draw, and self-enhance. arXiv preprint, 2024.
>
> >[4] Junfeng Wu, Yi Jiang, Chuofan Ma, Yuliang Liu, Hengshuang Zhao, Zehuan Yuan, Song Bai, and Xiang Bai. Liquid: Language models are scalable and unified multi-modal generators. arXiv preprint, 2024.
>
> >[5] Chenming Zhu, Tai Wang, Wenwei Zhang, Jiangmiao Pang, and Xihui Liu. Llava-3d: A simple yet effective pathway to empowering lmms with 3d-awareness. arXiv preprint, 2024a.
>
> >[6] Duo Zheng, Shijia Huang, and Liwei Wang. Video-3d llm: Learning position-aware video representation for 3d scene understanding. arXiv preprint, 2024.
>
> >[7] Dingning Liu, Xiaoshui Huang, Yuenan Hou, Zhihui Wang, Zhenfei Yin, Yongshun Gong, Peng Gao, and Wanli Ouyang. Uni3d-llm: Unifying point cloud perception, generation and editing with large language models. arXiv preprint, 2024.

---

> ### Comment · Reviewer_4To6 · 2025-11-26
>
> Dear authors,
>
> Thanks for your rebuttal!
>
> For the unification of 3D understanding and generation, I feel justified with the answers of the motivation of the unified model, and the results of jointly optimizing both generation and understanding only being slightly lower than separately optimizing them.
>
> For 3D benchmark evaluation, I think that this is why I feel the paper is overclaiming about solving 3D tasks. Spatial reasoning tasks, in my opinion, are somehow different from 3D understanding tasks, which is more like 2.5D scene understanding if we make an analogy to classical computer vision. And your new experimental results have shown that, without explicit 3D information, the 3D task performance will get inferior performance compared to existing methods, which further indicates that 3D tasks are more complicated than the spatial reasoning tasks benchmarked and focused in the paper.
>
> For the paper presentation, I appreciate the authors for revising the text and figures. Adding the Section 3.1 definitely help the understanding of the method. However, I still feel the figures are too overloaded and there are now as many as 4 figures in the method section explaining the method. My sense is, can the figures 3-5 be incorporated into a concise figure? Like Figure 4 could be the base one which we can revise on, as Figure 5 is only the inference version of Figure 4, and if I understand correctly, only the spatial decoder is the added module in Figure 5 compared to Figure 4. And the spatial decoder is obtained in the pretraining stage in Figure 3. So they should be able to be illustrated in one single figure. (Sorry for the long description of the figures, but the current set of figures is really that lengthy and still a bit hard to understand.)

---

> ### Author Response · Authors · 2025-11-27
>
> Dear Reviewer 4To6
>
> Many thanks for further response and continued engagement. We are pleased that **our response has addressed the reviewer’s concerns about the motivation for joint training and the performance comparison with separate optimization**. We also sincerely appreciate **the reviewer’s positive recognition of the motivation for the unified model**. For the 3D benchmark evaluation and paper presentation, we would like to provide further clarification and update the manuscript to address the reviewer's concerns.
>
> **3D benchmark evaluation**
>
> As the reviewer points out, when explicit 3D information is unavailable, our model underperforms 3D-enhanced methods on benchmarks that require precise spatial localization. We agree with this observation and would like to clarify our scope and contributions as follows:
>
> (1) Our objective is not to solve all 3D understanding tasks or to compete head-to-head with methods that operate on explicit 3D inputs (e.g., point clouds, meshes).
> Instead, our goal is to **explore a unified framework and training paradigm that enables a single model to support both spatial understanding and 3D-aware generation from RGB views**.
> In this sense, our focus is on expanding the scope of multimodal applications under 2D-only inputs, rather than claiming to replace specialized 3D understanding methods.
>
> (2) The complexity of spatial understanding does not stem solely from whether explicit 3D inputs are available, but also from the way tasks are formulated and supervised. In our setting, the model learns to **reason about geometry and spatial relationships from multi-view RGB images, without consuming ground-truth 3D representations** at test time. By contrast, many 3D benchmarks assume access to explicit geometric inputs (e.g., point clouds, depth maps), which define a different and complementary problem regime. We therefore position our work as focusing on a “multi-view spatial reasoning” setting, rather than claiming to cover the full spectrum of 3D understanding tasks.
>
> (3) The performance gap on 3D benchmarks can be attributed to two main factors:
>
> (i) Our model is **not pretrained on 3D data and does not take explicit 3D inputs**; it only receives multi-view 2D (or 2.5D) information during training. This naturally puts it at a disadvantage on benchmarks designed for 3D-enhanced models. At the same time, relying on explicit geometric inputs has its own **limitations**, such as the scarcity of large-scale real-world 3D datasets and the difficulty of fully leveraging powerful 2D pretrained encoders, which ultimately affects applicability and cross-task generalization. In contrast, **our framework reduces dependence on explicit 3D data** by learning geometry (e.g., depth and spatial relations) from multi-view images, which we believe is a more challenging yet scalable and promising direction.
>
> (ii) Part of the gap is due to **model capacity**. Our main experiments use a 3B backbone, whereas methods such as LLaVA-3D and Video-3D-LLM typically adopt 7B backbones. When we replace our 3B backbone with InternVL2.5-8B (still without 3D inputs), we obtain **competitive results**: SQA3D: 58.1 (EM); ScanQA: 15.3 (BLEU-4), 90.7 (CIDEr), 27.7 (EM); ScanRefer: 48.8 (Acc\@0.25), 43.1 (Acc\@0.5). This suggests that our framework has strong potential for spatial reasoning and grounding **without explicit 3D supervision**, while also supporting more complex generative tasks.
>
> In summary, we aim to explore **a unified framework for 3D-aware scene understanding and generation from RGB views**, rather than to claim that we solve all 3D understanding tasks. We are grateful to the reviewer for this feedback, which helps us clarify the motivation and limitations of our work. To avoid any potential over-claim, we have **explicitly discussed the performance limitations of our model on 3D benchmarks in Section 4.3** of the revised manuscript.
>
> **Paper presentation**
>
> Thanks for the reviewer's suggestion regarding the figure organization. As the reviewer pointed out, Figure 5 is essentially the inference version of Figure 4, and the spatial decoder is obtained in the pretraining stage illustrated in Figure 3. Following the reviewer's recommendation, we have **removed Figure 5** in the revised manuscript and integrated its content into Figure 4. The updated Figure 2 (d) and Figure  4 clearly present the inference workflow. This revision **simplifies figures** in the method section, reduces redundancy, and **improves the overall clarity** of the presentation.
>
> We deeply appreciate the reviewer's constructive engagement with our work and welcome any further questions or suggestions.
>
> Best wishes
>
> Authors

---

### Official Review · Reviewer_bejZ · 2025-11-01

**Soundness:** 3
**Presentation:** 3
**Contribution:** 3
**Rating:** 8
**Confidence:** 3

**Summary:**

The authors propose a unified framework for 3D understanding and generation, introducing a geometric–semantic learning strategy for encoder training based on the MASt3R architecture. The method includes a Spatial-VAE that compresses visual features into latent tokens for efficient 3D decoding, and a joint training setup where a large language model (LLM) and a diffusion model predict new views. Additionally, the model is trained with visual question answering (VQA) tasks to maintain its 3D understanding capability. Quantitative and qualitative experiments are conducted on both 3D generation and 3D understanding tasks across various spatial reasoning benchmarks.

**Strengths:**

1. Unifying 3D understanding and generation is an interesting and novel research direction.
2. The proposed geometry–semantic encoder is well-designed; incorporating the MASt3R framework is an effective and interesting choice.
3. The paper is well-written, and the experiments are thorough, covering a diverse set of novel view synthesis and vision–language spatial reasoning benchmarks.

**Weaknesses:**

1. The paper would benefit from a more detailed discussion of limitations and failure cases.
2. The method’s robustness to more extreme view transformations is not clearly evaluated.

**Questions:**

1. Could the authors elaborate on the main limitations and share examples of failure cases?
2. How sensitive is the model to large viewpoint changes in the 3D generation task?
3. I am also curious on whether joint training on understanding and 3D generation introduce any performance trade-offs between the two tasks.

---

> ### Author Response · Authors · 2025-11-22
> **Response to reviewer bejZ**
>
> Thanks for the positive and detailed review as well as the valuable suggestions for improvement. We would like to address the reviewer's concerns as follows:
>
> **W1: Lack of discussion of limitations and failure cases.**
>
> Thanks for the comment. Following the reviewer's suggestion, we have **added a discussion of the model’s limitations in Section 5** of the revised manuscript, including the lack of support for editing generated content and the inability to achieve controllable generation driven by language instructions.
>
> In addition, **we include several failure cases in Figures 12-13, and provide a corresponding discussion in Appendix A.3.3**. We hope these additions provide a clearer understanding of the limitations of our framework.
>
> **W2: Lack of viewpoint robustness evaluation.**
>
> Thanks for the suggestion. **We evaluate the generation under extreme view transformation in Appendix A.3.3. As shown in Figure 12**, we evaluate the model performance conditioned on rotation angles of 60°, 80°, 100°, 120°, and 140°.
> When the rotation angle of the view transformation is below 120°, our method can still produce semantically rich and structurally coherent 3D point clouds. However, **as the rotation angle increases to 140° or beyond, the generation quality drops noticeably**, with the point clouds exhibiting blurred textures and distorted structures.
>
> This degradation arises from two factors.
> First, during training, the view-transformation conditions are usually constrained within: view-overlap-ratio > 0.4, rel‑translation < 2 m, and rel‑rotation < 120°. Consequently, when evaluated under extreme view transformation, our model struggles to maintain high-quality generation. Second, such large viewpoint changes significantly reduce the overlap between the reference and target images, resulting in insufficient cross‑view constraints and ultimately leading to failures in 3D generation.
>
> **Q1: Limitations and failure cases.**
>
> Thanks for the question. In response to W1, we have added a discussion of the model’s limitations in **Section 5** and included representative failure cases in **Figures 12-13** of the revised manuscript.
>
> **Q2: Sensitivity to large viewpoint changes.**
>
> Thanks for the question. We evaluate the model’s robustness to large viewpoint changes in **Appendix A.3.3** and provide the corresponding visualizations in **Figure 12** (see response to W2).
>
> **Q3: Trade-offs in joint training.**
>
> Thanks for the question. We have added experiments that separately train the understanding task and the 3D generation task **in Appendix A.3.4 and Table 10. The results are reported as follows**:
>
> | Method    | ARKitScenes       |        |         | ScanNet++         |        |         |
> |--------------|:-------------------------------------:|:-----------------------------:|:-----------------------------:|:------------------------------------:|:-----------------------------:|:------------------------------------:|
> |    | FID↓ | KID↓   | LPIPS↓ |  FID↓ | KID↓   | LPIPS↓ |
> |CUT3R | 138.54 |0.1128 |0.5758 | 130.76 |0.1051 |0.5637  |
> |LVSM | 269.45 |0.3088 |0.5067 | 414.63 |0.5117 |0.5865 |
> | UniUGG (full) | 55.01   | 0.0425  | 0.4849  | 55.64      | **0.0442**  | 0.4263  |
> | **UniUGG (gen. only)** | **54.35**     | **0.0409**  | **0.4556**  |**55.46**       | 0.0454  | **0.3586**  |
>
> | Method    | VSI       |   BLINK     |     3DSR    | SPAR|
> |--------------|:-------------------------------------:|:-----------------------------:|:-----------------------------:|:------------------------------------:|
> | UniUGG (full) | 40.1| 43.6| **52.1**|47.2|
> | **UniUGG (und. only)**| **42.2**     | **44.4**  | 51.3  |**49.8** |
>
> Benefiting from the Spatial-VAE and diffusion model, **joint training does not significantly degrade the performance of 3D generation and only leads to a moderate reduction in spatial understanding performance**. Nevertheless, as shown in Table 3 in the revision, UniUGG still maintains competitive understanding performance compared with mainstream models.
>
> We believe **this performance trade-off is acceptable**: although a small amount of understanding performance is sacrificed, we obtain a unified 3D model for both understanding and generation. This unification substantially reduces training cost and computational overhead while broadening the range of tasks the model can support.

---

### Official Review · Reviewer_Qmfc · 2025-11-04

**Soundness:** 3
**Presentation:** 3
**Contribution:** 3
**Rating:** 8
**Confidence:** 3

**Summary:**

The paper proposes an LLM-based 3D framework for spatial-VQA and 3D scene generation. It addresses the limitations of current vision-language models, which are largely restricted to 2D semantics and next-token objectives that fail to capture spatial and geometric structure. To this end, the paper proposes a geometric-semantic pretraining strategy for the image encoder using multiview supervision. A spatial-VAE module then compresses the geometric-semantic features into compact 4-dimensional latent tokens for geometry-conditioned image synthesis. The model combines a geometry-aware encoder, a spatial-VAE model and a language conditioned diffusion generator. Experiments are done on Feat2GS benchmark for novel-view synthesis and several vision-language reasoning benchmarks.

**Strengths:**

1. The encoder is trained using geometric supervision which enables spatial reasoning compared to models trained on 2D images.
2. A single latent representation is used for both 3D reasoning and 3D generation.
3. The method allows geometric control through text or parametric view prompts.

**Weaknesses:**

1. There’s no discussion or visualization of the 3D output from the model.
2. All the evaluations are done in 2D space and there is no 3D evaluation performed. It’s unclear if the model is actually learning the structure of the 3D world or simply rendering based on the distribution of the training data.
3. The model is not trained end-to-end. The encoder and spatial VAE are trained separately and kept frozen during unified training (lines 301-302).

**Questions:**

1. There’s no discussion or evaluation of 3D output. Does the model learn explicit 3D geometry or only 3D features?
2. How does the model perform if the full model is trained end-to-end?
3. In the language-driven view transformation, how accurate is the angular or spatial accuracy?
4. What is represented by the 4-dimensional spatial-VAE tokens?

---

> ### Author Response · Authors · 2025-11-22
> **Response to reviewer Qmfc (1/3)**
>
> Thanks for the positive and detailed review as well as the valuable suggestions for improvement. We would like to address the reviewer's concerns as follows:
>
> **W1: Lack of 3D visualization.**
>
> Thanks for the comment. We would like to clarify that we do include discussions and visualizations of the generated 3D point clouds in our paper. In **Section 4.4** “Qualitative understanding and generation comparison”, we present and discuss 3D outputs of UniUGG, as shown in **Figure 6**, where the model generates 3D point clouds and performs captioning or visual question answering over original or generated scenes.
>
> In the paper, only Figure 5 shows 2D projections of the 3D outputs for better visualization comparison in the ablation study, highlighting the effects of different components and associated training paradigms. **Other visualizations—Figure 6 in the main text, and Figures 7–10 in the Appendix—are all explicit 3D point cloud visualizations**, not 2D projection renderings. In these visualizations, we qualitatively compare our method against baselines in terms of geometric structure and texture detail, and we also present various 3D scene generation results under different random seeds to demonstrate the diversity and superiority of our model’s output.
> Moreover, we **add additional 3D visualizations in Figures 12-13** during the rebuttal phase.
> If the reviewer is interested in other forms of 3D visualization, please feel free to let us know.
>
> **W2: Lack 3D evaluation.**
>
> Thanks for the reviewer's suggestions. We would like to clarify that UniUGG is designed for 3D generation rather than 3D reconstruction. Unlike reconstruction tasks, **the 3D generation field currently lacks universal and standardized evaluation metrics**, particularly for assessing generation quality and  **generation diversity**. For this reason, our original submission did not include 3D evaluation results. Instead, following common practice in image generation, we project the generated point clouds onto the image plane and conduct evaluation with standard image generation metrics, including Fréchet inception distance (FID), kernel inception distance (KID), and LPIPS (see Table 4 in the revision). These metrics are widely used to evaluate the quality of the generated distribution and perceptual similarity.
>
> **Following the reviewer's suggestion, we additionally conduct the 3D evaluation for the generated point clouds and report results as follows:**
>
>
> |   **Method**      | **Acc.[cm]↓**                       | **Comp.[cm]↓**              | **Chamfer dist. [cm]↓**      |
> |----------------|:-------------------------------------:|:-----------------------------:|:-----------------------------:|
> | CUT3R            |             14.388                        |          38.458                   |               26.423              |
> | *UniUGG (Ours)*  |                8.023                     |          24.082                   |              16.052               |
>
> We compute Accuracy(Acc.), Completeness(Comp.), and Chamfer Distance,
> which are commonly adopted in the 3D reconstruction field. Acc. and Comp. evaluate the precision and completeness of the generated point cloud, respectively, while Chamfer Distance measures the overall geometric discrepancy between the generated and ground-truth point clouds.
>
> Since LVSM[1] only renders novel-view images without explicitly generating 3D point clouds, we conduct 3D evaluation solely on UniUGG and CUT3R[2], both of which produce point clouds under novel view.
> As shown in the table, our method outperforms CUT3R across all evaluation metrics.
> Although these metrics are mainly used for the 3D reconstruction field,  they show that UniUGG can capture the structure of the 3D world rather than simply rendering images. We hope this can help address the reviewer's concern.

---

> ### Author Response · Authors · 2025-11-22
> **Response to reviewer Qmfc (2/3)**
>
> **W3: Not end-to-end trained.**
>
> Thanks for the comment. As the reviewer point out, our model is not trained end-to-end; instead, **we adopt a three-stage training strategy to address the core challenges of unified 3D modeling and improve the model's effectiveness.** As illustrated in Figure 2 of the revised manuscript, our training pipeline consists of three stages: (1) Pretrain the vision encoder to learn geometric-semantic representations, (2) Pretrain the Spatial-VAE to compress representations, and (3) Jointly train the UniUGG for spatial reasoning and 3D generation, with the vision encoder and VAE encoder frozen. As we claimed in the response to W1 of Reviewer D7LE, our motivation for this design is mainly due to two key challenges for unified 3D frameworks: (1) the limited capability of visual representations to capture both semantic and geometric information, and (2) the incompatibility between 3D generation and large language models.
>
> The pretraining of Spatial-VAE aims to compress and reconstruct spatial features. It is an existing standalone module and does not require adaptation to the LLM. Similarly, the vision encoder is pretrained only to extract geometry-semantic representations. **This modular training strategy is similar to many 2D unified models[3][4][5], which also rely on separately pretrained components (e.g., encoders, LLMs, or diffusion models) rather than full end-to-end optimization.** Moreover, end-to-end training would degrade performance (see response to Q2), making the three-stage training paradigm a more effective and stable solution for UniUGG.
>
> Notably, our work introduces the first unified architecture for spatial understanding and 3D generation. We aim to establish a new architectural paradigm that can inspire future research and serve as a foundation for more advanced unified 3D models.
>
> **Q1: Lack of 3D output analysis**
>
> Thanks for the question. Our model learns and generates explicit 3D geometry structures, rather than merely 3D features. We discussed and visualized the 3D outputs in the paper (see the response to W1). In addition, Following the reviewer's suggestion, we conducted 3D evaluations on the generated point clouds to further assess our UniUGG (see the response to W2). These results demonstrate that UniUGG can capture explicit 3D geometry.
>
> **Q2: End-to-end training performance**
>
> Thanks for the question. If the full model is trained end-to-end, the pipeline will necessarily be simplified in two aspects:
> (1) Pretraining of the geometry-semantic encoder (training stage 1) will be removed.  LLM will receive visual inputs from standard vision encoders, which are pretrained on 2D image semantic tasks.
> (2) Pretraining of the Spatial-VAE (training stage 2) will also be removed. It means utilize the diffusion model with the default VAE to generate spatial scenes.
>
> For simplification (1), we report the corresponding experimental results in Table 2 and Table 3. **The encoder directly trained in an end-to-end manner performs significantly worse than our pretrained geometry-semantic encoder in stage 1**, whether on spatial understanding or 3D generation tasks.
>
> For simplification (2), if we remove the pretrained Spatial-VAE and force the diffusion model to operate directly on high-dimensional features, the training becomes extremely unstable and fails to converge reliably. One could alternatively use a standard VAE and train it end-to-end. In fact, we have experimented with this strategy, but it similarly leads to **unstable optimization** and **produces features that cannot be decoded by the spatial decoder into semantically detailed and structurally coherent 3D point clouds**. Thus, as illustrated in Figure 4(a), our stage 2 training jointly optimizes the Spatial-VAE and the spatial decoder by reconstruction and spatial losses, ensuring that the generated features can be decoded into 3D point clouds.
>
> Overall, **fully end-to-end training substantially degrades performance on both spatial understanding and 3D generation**, whereas the proposed three-stage training paradigm yields a more stable and effective learning process.

---

> ### Author Response · Authors · 2025-11-22
> **Response to reviewer Qmfc (3/3)**
>
> **Q3: Language-driven view transformation accuracy**
>
> Thanks for the question. Our method **does not directly infer view transformation information from natural language**. Instead, we construct a Plücker ray map with pixel alignment based on the 4×4 view transformation matrix. This ray map is then encoded into queries using an MLP (lines 286–290 in the revision). The accuracy of this view transformation is supervised by the visual representation of the target image, allowing the model to achieve stable convergence in practice.
>
> Since the accuracy of the view-transformation condition cannot be quantitatively measured in a straightforward manner, we **provide qualitative visualization results in Appendix A.3.3, shown in Figure 11.**
> By taking only a reference image and the relative view transformation as input, UniUGG predicts the ViT tokens of the target image, which, together with the reference tokens, are then decoded into point clouds and the corresponding cross-view feature matchings.
> Our feature matching result is highly consistent with those produced by MASt3R, which takes image pairs as input.
> It demonstrates that the content generated by UniUGG is generally consistent with the input view transformation condition, which can reflect the accuracy of the view transformation.
>
> In future work, we will explore fully language-driven view transformations, enabling generation conditioned on textual instructions.
>
> **Q4: What is represented by the 4-dimensional spatial-VAE tokens?**
>
> Thanks for the question. Since the standard VAE is pretrained on 2D images, it cannot compress or reconstruct spatial-semantic representations. To align our framework with Stable Diffusion and support point-cloud synthesis, we pretrain the Spatial-VAE that compresses features into a compact 4-dimensional latent space. We consider these 4-dimensional Spatial-VAE tokens to be a compact and structurally meaningful visual embedding, which can be recovered into high-dimensional spatial features and finally decoded into 3D point clouds.
>
> >[1] Haian Jin, Hanwen Jiang, Hao Tan, Kai Zhang, Sai Bi, Tianyuan Zhang, Fujun Luan, Noah Snavely, and Zexiang Xu. Lvsm: A large view synthesis model with minimal 3d inductive bias. In ICLR, 2025.
>
> >[2] Qianqian Wang, Yifei Zhang, Aleksander Holynski, Alexei A Efros, and Angjoo Kanazawa. Continuous 3d perception model with persistent state. In CVPR, 2025b.
>
> >[3] Chengyue Wu, Xiaokang Chen, Zhiyu Wu, Yiyang Ma, Xingchao Liu, Zizheng Pan, Wen Liu, Zhenda Xie, Xingkai Yu, Chong Ruan, et al. Janus: Decoupling visual encoding for unified multimodal understanding and generation. In CVPR, 2025.
>
> >[4] Jinheng Xie, Weijia Mao, Zechen Bai, David Junhao Zhang, Weihao Wang, Kevin Qinghong Lin, Yuchao Gu, Zhijie Chen, Zhenheng Yang, and Mike Zheng Shou. Show-o: One single transformer to unify multimodal understanding and generation. In ICLR, 2025
>
> >[5] Chunwei Wang, Guansong Lu, Junwei Yang, Runhui Huang, Jianhua Han, Lu Hou, Wei Zhang, and Hang Xu. Illume: Illuminating your llms to see, draw, and self-enhance. arXiv preprint, 2024.

---

### Official Review · Reviewer_D7LE · 2025-11-05

**Soundness:** 1
**Presentation:** 1
**Contribution:** 4
**Rating:** 2
**Confidence:** 2

**Summary:**

The paper proposes a method to ingest images and can generate extended scenes based on text input or camera parameters. How this works is unclear to me. I've read the paper 3 times and I still don't fully understand how the different parts work together.

**Strengths:**

- **S.1:** The qualitative results look great.

**Weaknesses:**

- **W.1:** I've worked with LLMs, I've worked on diffusion papers, and I know about the Mast3r paper. Yet, I have no clue what's going on in this work and why. I think the writing is a bit too dense to follow unless the reader is already intimately familiar with all the related works. And I also don't think the figures are doing a great job at illustrating the method. I can't say anything more about this work. Maybe an overarching, vastly simplified diagram would be helpful.

**Questions:**

- **Q.1:** Doesn't the use of a VAE in the second step defeat the purpose of semantically aligning the encoder in the first step?
- **Q.2:** Why isn't the LLM trained directly to output tokens that can be decoded into visual representations by the VAE decoder, rather than predicting something that can be rendered through a diffusion process?

---

> ### Author Response · Authors · 2025-11-22
> **Response to reviewer D7LE (1/2)**
>
> Thanks for the positive and detailed review as well as the valuable suggestions for improvement. We would like to address the reviewer's concerns as follows:
>
>
> **W1: Presentation clarity concerns.**
>
> We understand that our work introduces multiple components (including geometry-semantic alignment, large language models, and the diffusion module), which may introduce some barriers to accessibility.
> Following the reviewer's suggestion, we have **added Section 3.1** in the revised manuscript to provide a clearer and more structured introduction to the overall workflow of our proposed UniUGG. We have also **included an overarching and highly simplified overview diagram** (Figure 2 in the revision) to help readers better grasp the key components and the overall pipeline. Additionally, we refined other figures in the revised manuscript by enlarging font sizes and addressing ambiguities to improve overall readability.
>
> We would also like to take this opportunity to clarify the motivation and key contributions of our work.
> Our goal is to build **a unified 3D framework for spatial understanding and generation**, analogous to existing unified 2D models. However, two key challenges currently hinder the development of such 3D frameworks:
> (1) the limited capability of visual representations to capture both semantic and geometric information, and
> (2) the incompatibility between 3D generation and large language models.
> To address the first issue, we introduce a geometric-semantic pretraining strategy for the vision encoder, designed to extract joint semantic and geometric visual representations. To address the second, we propose UniUGG, a framework that leverages a large language model together with a diffusion U-Net to jointly support spatial understanding and 3D generation.
> To further enhance generation quality and efficiency, we incorporate a Spatial-VAE module that compresses the geometric-semantic representations and produces sharper 3D point clouds. Thus, our training pipeline follows a **three-stage strategy**: (1) Pretrain the vision encoder to learn geometric-semantic representations, (2) Pretrain the Spatial-VAE to compress representations, and (3) Jointly train the UniUGG for spatial reasoning and 3D generation, with the vision encoder and VAE encoder frozen. As a result, we propose the **first LLM-based framework that unifies spatial understanding and 3D generation**. UniUGG enables both spatial-level visual question answering (VQA) and the generation of geometrically consistent 3D scenes from language and image inputs. Moreover, our method achieves top performance on multiple spatial reasoning benchmarks.

---

> ### Author Response · Authors · 2025-11-22
> **Response to reviewer D7LE (2/2)**
>
> **Q1: Impact of VAE on semantic alignment.**
>
> We appreciate the reviewer’s insightful question. We would like to clarify that the use of the **Spatial-VAE in our framework does not defeat the purpose of semantically aligning the vision encoder** in the first step.
> The purpose of semantic alignment is to ensure that the vision encoder produces both geometry-aware and semantically coherent visual representations for LLMs.
> The Spatial-VAE aims to compress visual representations, allowing the diffusion model to operate in a compact latent space rather than directly on high-dimensional features.
> The Spatial-VAE does not defeat the semantic alignment established in the encoder for the following reasons:
>
> (1) First, **during the training of the Spatial-VAE or UniUGG, the ViT encoder is always frozen.**
> We neither jointly train the Spatial-VAE with the encoder nor jointly optimize the LLM with the encoder. Consequently, the geometric–semantic features learned in the first stage remain fully preserved throughout the entire pipeline. This is also one of the motivations for adopting a three-stage training paradigm.
>
> (2) Second, as shown in Figure 4 (a) of the revised manuscript, **the Spatial-VAE is trained jointly with the spatial decoder of the first step.** This joint training ensures that the visual representations compressed and reconstructed by the Spatial-VAE preserve the geometric and semantic information learned during the first stage, enabling the spatial decoder to recover the corresponding 3D point clouds accurately.
>
> (3) Third, in the final UniUGG framework (Figure 4(b) in the revision), the pretrained ViT encoder (frozen) is used to encode images into geometry-semantic representations, which are directly fed into the LLM without any compression by the Spatial-VAE. It means that the **LLM receives lossless, semantically aligned visual features** to output answer tokens or conditional features, which are further used for 3D understanding and generation tasks.
>
> (4) Finally, **our experiments demonstrate the effectiveness of this design**. As shown in Table 3 (right) of the revised manuscript, the diffusion module, along with the Spatial-VAE, significantly improves 3D generation quality. Additionally, our ViT encoder achieves superior performance in both understanding and generation compared to other encoders (Table 2 and Figure 5).  These results show that the semantic alignment from the first step is preserved and leveraged throughout the pipeline.
>
> Therefore, the Spatial-VAE improves efficiency for latent-space diffusion without affecting the semantic alignment used by the LLM.
>
>
> **Q2: Why choose a diffusion process rather than direct LLM output?**
>
> Thanks for pointing out this insightful question. In summary, diffusion-based generation yields better 3D quality than direct LLM-to-token decoding, as confirmed by our experiments.
>
> As the review suggested, our initial idea was indeed to have the LLM directly predict visual tokens that could be decoded into 3D point clouds, thus eliminating the need for a diffusion process. In that case, the Spatial-VAE would also be unnecessary, since it was designed for latent space diffusion. **As discussed in Appendix A.4, we experimented with this idea by having the LLM directly generate ViT visual tokens**, which were then decoded by the spatial decoder into point clouds (see the pipeline in Figure 14). **While this design could capture spatial structure under novel views, the resulting point clouds lacked details and textures** (see results in Figure 15). Directly predicting target-view representations (either high-dimensional Z or low-dimensional latents) is costly and unstable, and tends to average over the inherently multi-modal novel-view distribution, leading to blurred geometry and missing fine details. To address this limitation, we introduced a diffusion process in UniUGG to enhance the generation of sharper, more detailed 3D outputs.
>
> Our experimental results also validate the effectiveness of this design. **As shown in Table 3 (right) and Figure 5, removing the diffusion process(“w/o Diffusion”) leads to a noticeable drop in generation quality**.

---

> > ### Comment · Reviewer_D7LE · 2025-11-25
> >
> > Thanks to the authors for their responses and for the vastly improved new figure and the great overview section. I've updated my rating.

---

> > > ### Author Response · Authors · 2025-11-26
> > >
> > > Dear Reviewer D7LE
> > >
> > > Many thanks for the positive feedback and support. We appreciate the reviewer's time for reviewing and thanks again for the valuable comments.
> > >
> > > Best wishes
> > >
> > > Authors

---

### Author Response · Authors · 2025-11-22
**Summary of responses to all reviewers and ACs**

We thank all the reviewers for their insightful comments and the AC for the extra effort. We are encouraged that our unified framework for 3D understanding and generation is recognized as novel and pioneering, which can serve as a foundational work for future unified 3D models (bejZ, 4To6). Reviewers highlighted that the pretraining of our geometry–semantic encoder and spatial decoder is innovative, well-designed, and effective (Qmfc, bejZ, 4To6), and further noted that leveraging Spatial-VAE for latent generation constitutes a meaningful contribution (Qmfc). Additionally, the proposed method is praised for its great qualitative results and well-rounded performance across a diverse set of benchmarks, including visual representation and spatial understanding, while also enabling geometry-aware controllable generation through parametric viewpoint prompts (D7LE, Qmfc, bejZ, 4To6).

---
In response to the reviewers’ comments, we have revised the paper accordingly and summarize the major updates as follows:

- We add Section 3.1 and include an overview diagram in Figure 2 to provide a clearer and more structured introduction to the overall workflow of our model. (D7LE, 4To6)

- We refine Figures 3-4 by improving the layout, correcting errors, and enlarging the font sizes to enhance readability. We also removed original Figure 5 to reduce redundancy and improve the overall clarity of the presentation. (D7LE, 4To6)

- We report the training cost and computational resources in Section 4.1, “Implementation Details”. (4To6)

- A discussion of our model’s limitations is provided in Section 5. (bejZ)

- The comparison and discussion of the model’s performance on the SQA3D, ScanQA, and ScanRefer benchmarks are provided in Table 9 and Appendix A.3.2. (4To6)

- We add the “Feature matching visualization,” “Robust generation under extreme view transformation,” and “Failure cases” sections in Appendix A.3.3, “Evaluation of 3D Generation”. The corresponding experimental results are presented in Figures 11–13. (Qmfc, bejZ)

- We include experiments that separately optimize the spatial understanding and 3D generation tasks, and compare their performance with that of the jointly trained model in Appendix A.3.4 and Table 10. (bejZ, 4To6)

- We highlight all modifications in the manuscript.

The other concerns raised by the reviewers have also been addressed individually.

In the early stage of the rebuttal, **our response effectively addressed the concerns raised by Reviewer D7LE, resulting in a score increase from 2 to 6**.
After our first response, **Reviewer 4To6 also acknowledged that the concerns regarding “the motivation for the unified model and the comparison with separate optimization” were resolved.** For the reviewer’s subsequent comments on the 3D benchmark evaluation and the paper presentation, we provided further clarification and updated the manuscript accordingly to address all remaining concerns.

---
Here, we conclude our main contributions as follows:
- We propose **the first unified framework for spatial understanding and 3D generation**, enabling spatial-level VQA and producing geometrically consistent, richly detailed 3D scenes.
- We introduce **a novel geometry–semantic encoder pretraining strategy**, in which our encoder extracts geometric cues from images while preserving semantic features from 2D priors.
- We introduce **the Spatial-VAE module to enable efficient and high-quality 3D generation**. It compresses the geometry–semantic representations, allowing the diffusion model to operate in a compact latent space rather than directly on high-dimensional features.
- Our method achieves **top performance** on multiple spatial reasoning benchmarks and maintains **significant superiority** in 3D generation.

---

We believe the additional experiments solidly validated our claims. We appreciate the opportunity to further improve our work and hope the AC finds these revisions satisfactory. Thanks once again to the reviewers and the AC for their valuable contributions to the community.

---

### Meta-Review · Area_Chair_ajFd · 2025-12-16

**Summary:**

The paper aims at developing a unified model for both 3D understanding and generation with both geometric and semantic information within input images are encoded with the visual encoder and a spatial decoder for 3D generation.

Strengths identified by reviewers include, its novel and interesting research direction, well model designs, well-writing and thorough experiments.

The reviewers raise several concerns and the major concern is that the presentation of the paper (e.g., figures are unclear and confusing, the writing of the methodology section is difficult to follow) is unclear and confusing. Additionally, reviewers also have concerns on the motivation of unifying both 3D understanding and generation, the evaluation on 3D understanding, and more discussion on failure cases and robustness.

Reviewer D7LE initially suggested negative score due to the unclear presentation clarity and has confirmed raising the scores to positive as the authors have provided detailed feedback and revised the manuscript to positive score. Given that two other reviewers recommended positive scores and most of concerns have been addressed, the AC recommended acceptance of the paper.

**Reviewer Concerns:**

The authors have provided very detailed responses with additional results and revised the manuscripts accordingly. This has addressed most concerns, including, 1) presentation clarity; impact of VAE on semantic alignment; 2) choice of diffusion process compared to LLM output; 3) lack of discussion or visualization of the 3D output from the model; 4) lack of 3D evaluation; 5) results of end-to-end training; 6) accuracy of angular and spatial in language-driven view transformation; 7) 4d spatial-VAE tokens; 8) detailed discussion of limitations and failure cases; 9) robustness to more extreme view transformations; 10) joint training of both tasks; 11) motivation for unifying both tasks; 12) lack of overview at the beginning of the methodology section; 13) training cost analysis.

**Reviewer Scores:**

Initially, the reviewer scores are mixed (two accepts, one borderline reject and one reject). Reviewer D7LE has confirmed raising the score to 6 and other reviewers would have maintained or raised their scores as most of concerns have been addressed.

---

### Decision · Program_Chairs · 2026-01-26

Accept (Poster)